# Repression of ferritin light chain translation by human eIF3

**Mia C Pulos-Holmes[1], Daniel N Srole[1†], Maria G Juarez[1], Amy S-Y Lee[2], David T McSwiggen[1], Nicholas T Ingolia[1,3], Jamie H Cate[1,3,4,5]***

[1]Department of Molecular & Cell Biology, University of California, Berkeley, Berkeley, United States; [2]Biology Department, Rosenstiel Basic Medical Science Research Center, Brandeis University, Waltham, United States; [3]California Institute for Quantitative Biosciences, University of California, Berkeley, Berkeley, United States; [4]Department of Chemistry, University of California, Berkeley, Berkeley, United States; [5]Molecular Biophysics & Integrated Bioimaging Division, Lawrence Berkeley National Laboratory, Berkeley, United States

**Abstract** A central problem in human biology remains the discovery of causal molecular links between mutations identified in genome-wide association studies (GWAS) and their corresponding disease traits. This challenge is magnified for variants residing in non-coding regions of the genome. Single-nucleotide polymorphisms (SNPs) in the 5' untranslated region (5'-UTR) of the ferritin light chain (*FTL*) gene that cause hyperferritinemia are reported to disrupt translation repression by altering iron regulatory protein (IRP) interactions with the *FTL* mRNA 5'-UTR. Here, we show that human eukaryotic translation initiation factor 3 (eIF3) acts as a distinct repressor of *FTL* mRNA translation, and eIF3-mediated *FTL* repression is disrupted by a subset of SNPs in *FTL* that cause hyperferritinemia. These results identify a direct role for eIF3-mediated translational control in a specific human disease.
DOI: https://doi.org/10.7554/eLife.48193.001

*For correspondence:
jcate@lbl.gov

Present address: †Department of Molecular & Medical Pharmacology, David Geffen School of Medicine at UCLA, University of California, Los Angeles, Los Angeles, United States

Competing interests: The authors declare that no competing interests exist.

## Introduction

Iron is essential for a spectrum of metabolic pathways and cellular growth. However, if not properly managed, excess iron catalyzes the production of reactive oxygen species (ROS). To safeguard against these toxic effects, cells sequester iron in ferritin, a cage-like protein complex composed of a variable mixture of two structurally similar but functionally distinct subunits, the ferritin heavy chain (FTH) and the ferritin light chain (FTL) (*Harrison and Arosio, 1996*), (*Knovich et al., 2009*). To maintain iron homeostasis, the expression of both ferritin subunits in mammals is regulated post-transcriptionally by iron regulatory proteins that bind a highly conserved RNA hairpin called the iron responsive element (IRE), located in the 5'-UTRs of *FTL* and *FTH1* mRNAs (*Figure 1A and B*) (*Theil, 1994*), (*Wilkinson and Pantopoulos, 2014*). SNPs or deletions in the 5'-UTR that disrupt IRP-IRE interactions are thought to be the primary cause of hereditary hyperferritinemia cataract syndrome, a condition involving an abnormal buildup of serum ferritin in the absence of iron overload (*Cazzola et al., 1997*).

Although the IRP-IRE interactions have been considered to be the sole post-transcriptional means of regulating ferritin expression, recent studies have provided strong evidence that other presently unknown factors may provide another layer of regulation during *FTL* translation. For example, the FTL subunit composition of ferritin is altered in response to environmental factors such as hypoxia (*Sammarco et al., 2008*). We recently found that eIF3 can function beyond its scaffolding role in general translation initiation by acting as either an activator or repressor of translation in a transcript-specific manner (*Lee et al., 2015*),(*Lee et al., 2016*). This regulation occurs through

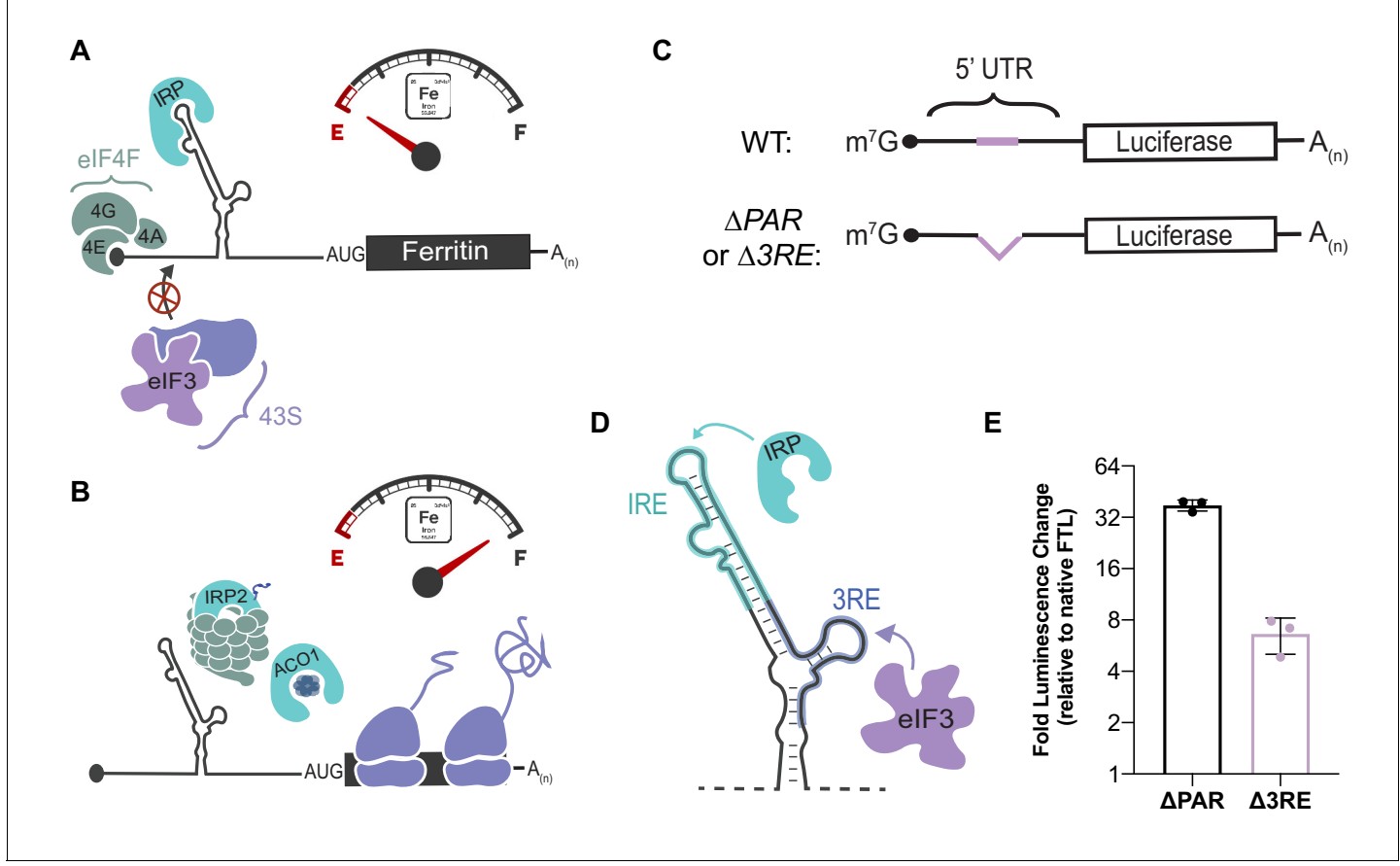

**Figure 1.** Post-transcriptional regulation of *FTL* mRNA. (**A, B**) Iron-responsive regulation mediated by binding of Iron Response Proteins (IRPs) to Iron Response Element (IRE) RNA structures in the 5'-UTR in (**A**) low-iron conditions and (**B**) high-iron conditions. In high iron, IRP2 is degraded by the proteasome, whereas IRP1 binds an iron-sulfur cluster to form the enzyme Aconitase (ACO1). (**C**) General schematic of the luciferase reporter mRNAs. The eIF3 PAR-CLIP site in *FTL* mRNA spans nucleotides 53–76 (*Lee et al., 2015*) and the 3RE region spans nucleotides 58–90. (**D**) Schematic of the IRP and eIF3 interaction sites on the experimentally-determined secondary structure of *FTL* mRNA (*Martin et al., 2012*). (**E**) Luciferase activity in HepG2 cells transfected with luciferase reporter mRNAs 6 hr post transfection, normalized to luciferase luminescence from mRNA with wild-type *FTL* 5'-UTR. The results are for three biological replicates with error bars representing the standard deviation of the mean.

DOI: https://doi.org/10.7554/eLife.48193.002

The following source data and figure supplements are available for figure 1:

**Source data 1.** Luciferase reporter readouts.
DOI: https://doi.org/10.7554/eLife.48193.005

**Figure supplement 1.** Sites of eIF3 interaction with *FTL* and *FTH1* mRNAs.
DOI: https://doi.org/10.7554/eLife.48193.003

**Figure supplement 1—source data 1.** Luciferase reporter readouts.
DOI: https://doi.org/10.7554/eLife.48193.004

interactions with primarily 5'-UTR RNA structural elements (*Lee et al., 2015*). Notably, we found that *FTL* mRNA cross-links to eIF3 (*Lee et al., 2015*), but the role eIF3 plays in regulating *FTL* translation has not been established.

Here, we report a previously unknown mode of *FTL* translation regulation with a direct link to disease-related genetic mutations. We show that eIF3 binds to human *FTL* mRNA by means of sequences in the 5'-UTR immediately adjacent to the IRE, and provides additional regulation of *FTL* translation independent of IRP-IRE. After using CRISPR-Cas9 genome editing to delete the endogenous eIF3 interaction site in *FTL*, we monitored direct phenotypic responses of cells under normal and iron modulated conditions. Lastly, we used competitive IRP binding assays to explore the potential role of eIF3 in hyperferritinemia. These experiments reveal that eIF3 acts as a repressor of *FTL*

translation, and disruption of eIF3 interactions with *FTL* mRNA due to specific SNPs in the *FTL* 5'-UTR likely contributes to a subset of hyperferritinemia cases.

## Results

### Identification of the eIF3-*FTL* mRNA interaction site

In order to understand the functional effect of the interaction between eIF3 and *FTL* mRNA, we utilized *Renilla* luciferase (rLuc) reporter mRNAs in which the 5'-UTR from *FTL* was placed upstream of the *Renilla* coding sequence (*Figure 1C*). To measure the importance of the *FTL* mRNA region identified by PAR-CLIP (*Lee et al., 2015*), various mutations were introduced into the *FTL* 5'-UTR to disrupt eIF3 binding. The eIF3 binding site on the 5'-UTR of *FTL*, as determined by PAR-CLIP, spans a 24 nucleotide sequence that overlaps with the last five nucleotides of the annotated sequence of the *FTL* IRE (*Figure 1—figure supplement 1*). Notably, no eIF3 cross-linking site was observed in the 5'-UTR of the mRNA encoding *FTH1*, which shares the structurally conserved IRE, but not adjacent sequence features (*Figure 1—figure supplement 1B*) (*7*). The removal of the eIF3 interaction site dramatically increased translation of rLuc when compared to the full length wild type *FTL* 5'-UTR, 38-fold when the PAR-CLIP defined sequence was deleted (ΔPAR, nucleotides 53–76) and six fold in a deletion that maintained the full IRE sequence (eIF3 repressive element, Δ3RE, nucleotides 58–90) (*Figure 1D and E*, *Figure 1—figure supplement 1D*) (*Theil, 2015*). These results suggest that eIF3 binding to the *FTL* 5'-UTR represses *FTL* translation.

### Decoupling the repressive role of eIF3 on *FTL* mRNA from that of IRP

Due to the close proximity between the eIF3 interaction site and the *FTL* IRE, accompanied by the fact that the 5'-UTR of *FTL* is prone to large-scale structural rearrangements (*Martin et al., 2012*), we tested whether the derepression observed in the ΔPAR and Δ3RE mRNAs is a direct result of altering eIF3 binding and not due to disrupting the IRE-IRP interaction. To evaluate the effect of the deletions (ΔPAR, Δ3RE) on IRP binding, we carried out RNA-electrophoretic mobility shift assays with near-IR-labeled *FTL* 5'-UTR RNA and recombinant IRP1. As expected, IRP1 bound to the wild type 5'-UTR of *FTL* (*Figure 2A*). IRP1 also bound the Δ3RE RNA, but failed to bind efficiently to the ΔPAR RNA (*Figure 2A*). The loss of IRP1 binding to ΔPAR RNA could be attributed to disrupted RNA folding, or to the importance of the region of overlap between the IRE and the PAR-CLIP defined eIF3-binding site (*Figure 1—figure supplement 1C*). We further quantified IRP binding to the Δ3RE 5'-UTR using RNA binding competition assays (*Figure 2B and C*, *Figure 2—figure supplement 1*). IRP1 had only slightly attenuated binding to the Δ3RE 5'-UTR RNA when compared to the wild-type *FTL* 5'-UTR, suggesting that the alleviation of repression observed in the luciferase translation experiments for the Δ3RE mRNA (*Figure 1E*, *Figure 1—figure supplement 1D*) could be due to disruption of eIF3 binding.

To further ensure that the alleviation of repression of *FTL* translation by the Δ3RE mutation results primarily from disruption of eIF3-*FTL* binding and not IRE-IRP interactions, we modulated the location of the IRE in the 5'-UTR of *FTL*. It had been shown previously that moving the IRE in the *FTH1* 5'-UTR further than 60 nucleotides from the 5' $m^7$G-cap partially relieves IRP-dependent inhibition of *FTH1* translation (*Goossen and Hentze, 1992*). Inhibition is lost because bound IRP can no longer fully sterically block the assembly of the 43S preinitiation complex on the mRNA (*Muckenthaler et al., 1998*),(*Paraskeva et al., 1999*). To further investigate if this holds for *FTL* mRNA, we used the *FTL* and Δ3RE luciferase reporter constructs and placed the characteristic C bulge of the IRE either 32 nucleotides away (native) or 70 nucleotides away (extended) from the 5'-cap (*Figure 2D*). As with *FTH1* mRNA (*Goossen and Hentze, 1992*), moving the IRE further from the 5'-cap partially derepressed translation of the *FTL*-based luciferase reporter (*Figure 2E*). Notably, overall translation was much higher from both Δ3RE mRNAs, even with partial removal of IRP-dependent repression due to the distance from the 5'-cap (*Figure 2E*). This cap position-independent derepression is unique to *FTL* as the 3RE region is not conserved in the *FTH1* 5'-UTR. Furthermore, combining the Δ3RE mutation with a mutation that disrupts IRP binding to the IRE entirely (Loop) (*Cazzola et al., 1997*) synergistically derepressed luciferase reporter translation (*Figure 2F*, compare Double to Loop and Δ3RE). We verified the mRNAs were equally stable during the 6 hr time courses (*Figure 2—figure supplement 2*), ensuring the observed phenotypic changes are

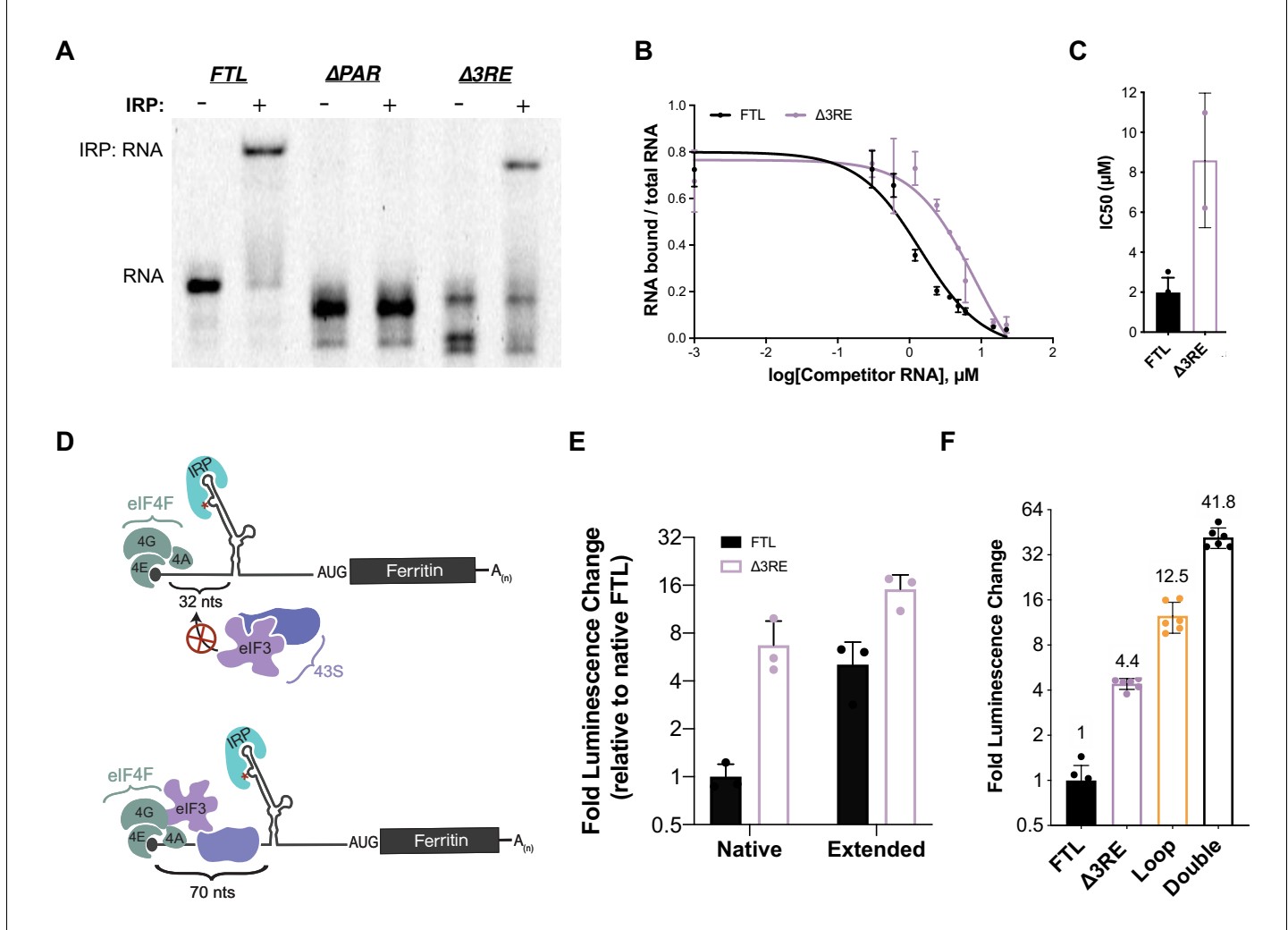

**Figure 2.** Maintenance of 5'-cap and IRP-dependent regulation of 3RE deletions in the *FTL* 5'-UTR. (**A**) Representative native RNA gel shift showing recombinant IRP1 binding activity. Near IR (NIR) labeled RNAs corresponding to the full-length WT *FTL* 5'-UTR and the *FTL* 5'-UTR with deletions of the predicted eIF3 interaction site were incubated with recombinant IRP1 and resolved on a native polyacrylamide gel. (**B**) Dose-response curve of two RNA competition assays, based on gel shifts of NIR-labeled WT IRE-containing RNA, with WT or Δ3RE RNAs serving as cold competitors. Fold excess of competitors extended up to 75,000x. Error bars represent standard deviations for each concentration of competitor. (**C**) Calculated $IC_{50}$ values using Prism 7 of the various competitor RNAs, based on the data in (**B**), with error bars representing the standard deviation from the mean $IC_{50}$ value. N.A., the $IC_{50}$ value could not be determined for the Loop mutant due to lack of any detectable competition. (**D**) Schematics showing the effect of increasing the distance of the IRE from the 5'-cap on IRP regulation of translation initiation (*Goossen and Hentze, 1992*), (*Muckenthaler et al., 1998*). The characteristic C (C18 in the wild-type context) is denoted by an asterisk. (**E**) The luciferase activity of HepG2 cells transfected with mRNAs containing the native and extended spacing between the 5'-cap and IRE, with or without the 3RE site, normalized to the luciferase luminescence of cells transfected with WT *FTL* mRNA with native spacing from the 5'-cap. The values are from cells that have been harvested 6 hr post-transfection. The results are from three biological replicates, with error bars representing the standard deviation of the mean. (**F**) The luciferase activity of HepG2 cells transfected with mRNAs containing the native and various combinations of eIF3 (Δ3RE) and IRP (Loop, A15G/G16C) disrupting mutations in HepG2 cells, normalized to the luciferase luminescence of cells transfected with WT *FTL* mRNA with native spacing from the 5'-cap. Double represents an mRNA construct that contains both the Loop and Δ3RE mutations. The results are from six independent transfections, with error bars representing the standard deviation of the mean.

DOI: https://doi.org/10.7554/eLife.48193.006

The following source data and figure supplements are available for figure 2:

**Source data 1.** EMSA analysis and luciferase reporter readouts.
DOI: https://doi.org/10.7554/eLife.48193.011
**Figure supplement 1.** Native gels resolving RNA-IRP1 complexes formed after competition experiments.
DOI: https://doi.org/10.7554/eLife.48193.007
*Figure 2 continued on next page*

*Figure 2 continued*

**Figure supplement 2.** Luciferase readouts and mRNA stability during 6 hr mRNA transfections.
DOI: https://doi.org/10.7554/eLife.48193.008
**Figure supplement 2—source data 1.** Luciferase reporter and qPCR readouts relative to native*FTL*.
DOI: https://doi.org/10.7554/eLife.48193.009
**Figure supplement 3.** Mathematical modeling of IRP and eIF3 co occupancy on *FTL* mRNA.
DOI: https://doi.org/10.7554/eLife.48193.010

caused by changes in translational regulation. The slight inconsistency in the amount of derepression observed with the double mutation (*Figure 2F*) and the extended Δ3RE mRNA (*Figure 2E*) may be due to the fact that the extension of the 5'-UTR from its native state does not completely abolish the repression of *FTL* mRNA translation by IRP, in contrast to the loop mutation which abolishes IRP binding (*Cazzola et al., 1997*). Furthermore, it is not clear how the distance from the 5'-cap affects repression mediated by eIF3. Taken together, these results indicate that IRP-mediated inhibition of translation is maintained in the Δ3RE mRNA, and that eIF3 confers an additional level of repression beyond that which can be provided by IRP.

We used the luciferase reporter results in *Figure 2F* to formulate a mathematical model to determine whether eIF3 and IRP can bind and regulate *FTL* mRNA simultaneously (See Materials and methods for details). Such a model would be useful in conditions with different extents of iron-response regulation. We defined a system in which IRP and eIF3 do not bind the same mRNA. This leaves three possible states in which the *FTL* mRNA could exist: the fraction bound solely by IRP ($x_1$), the fraction bound solely by eIF3 ($x_2$), and the remainder of the mRNA which is unbound by either factor ($x_3$) (*Figure 2—figure supplement 3*). We further elaborated this model to include translation efficiency ($y$). Here we assume the mutations do not affect the translation efficiency of the unbound species ($x_3$), while the other two populations ($x_1$ and $x_2$) have translational efficiencies ($y_1$ and $y_2$) scaled between full repression ($y_n = 0$) and no repression ($y_n = 1$). Lastly, we assumed the mutations that disrupt binding shift the equilibrium of the total mRNA population between bound and unbound fractions of the alternate factor by some amount, either $\alpha$ for those affecting eIF3 binding or $\beta$ for IRP binding. Using the translational output determined in *Figure 2F*, we find that the data are inconsistent with the model that eIF3 and IRP cannot bind the same mRNA. Rather, the data indicate that IRP and eIF3 likely act in cis on at least a fraction of the *FTL* mRNA.

## Physiological response to loss of eIF3-dependent repression

To investigate the physiological response to the loss of eIF3-based repression, we genetically engineered either HEK293T or HepG2 cells to generate the Δ3RE mutation in the 5'-UTR in the *FTL* gene (*Figure 3—figure supplement 1A–1C*). Notably, we found that the Δ3RE mutation abolished the preferential interaction of eIF3 with *FTL* mRNA (*Figure 3E,F*). Furthermore, FTL protein production increased dramatically in the Δ3RE cell lines, as expected from removing the predicted eIF3 repressive element (*Figure 3A*, *Figure 3—figure supplements 1D*, *4A* and *5A*). Importantly, the increase in FTL protein levels was not due to increases in mRNA levels (*Figure 3—figure supplement 2A*), indicating that the de-repression occurs post-transcriptionally. Interestingly, the increase in FTL levels occurs with a concurrent reduction in FTH levels (*Figure 3B*, *Figure 3—figure supplements 4C* and *5A*). The decrease in FTH protein levels is also not due to changes in *FTH1* mRNA levels (*Figure 3—figure supplement 2B*).

To test whether IRP maintains its ability to dynamically regulate *FTL* translation, we treated the cell lines with either ferric ammonium citrate (FAC), an iron donor, or desferoxamine (DFO), an iron chelator, to increase or decrease iron levels, respectively. (*Figure 3—figure supplement 3*, *Figure 3—figure supplement 4B and D*, *Figure 3—figure supplement 5A and B*) (*Schneider and Leibold, 2003*). FTL levels in the Δ3RE cell lines responded to FAC and DFO treatment in a comparable manner to the unedited (WT) cell lines (*Figure 3C–D*, *Figure 3—figure supplement 4B and D*, *Figure 3—figure supplement 5A and B*), showing that the Δ3RE transcript retains IRP-dependent translational regulation in cells. This iron-responsive regulation is maintained even though the basal levels of FTL protein are much higher in the Δ3RE compared to WT cells (*Figure 3D*). FTH levels also respond to iron levels in both the WT and Δ3RE cells (*Figure 3—figure supplement 3*, *Figure 3—*

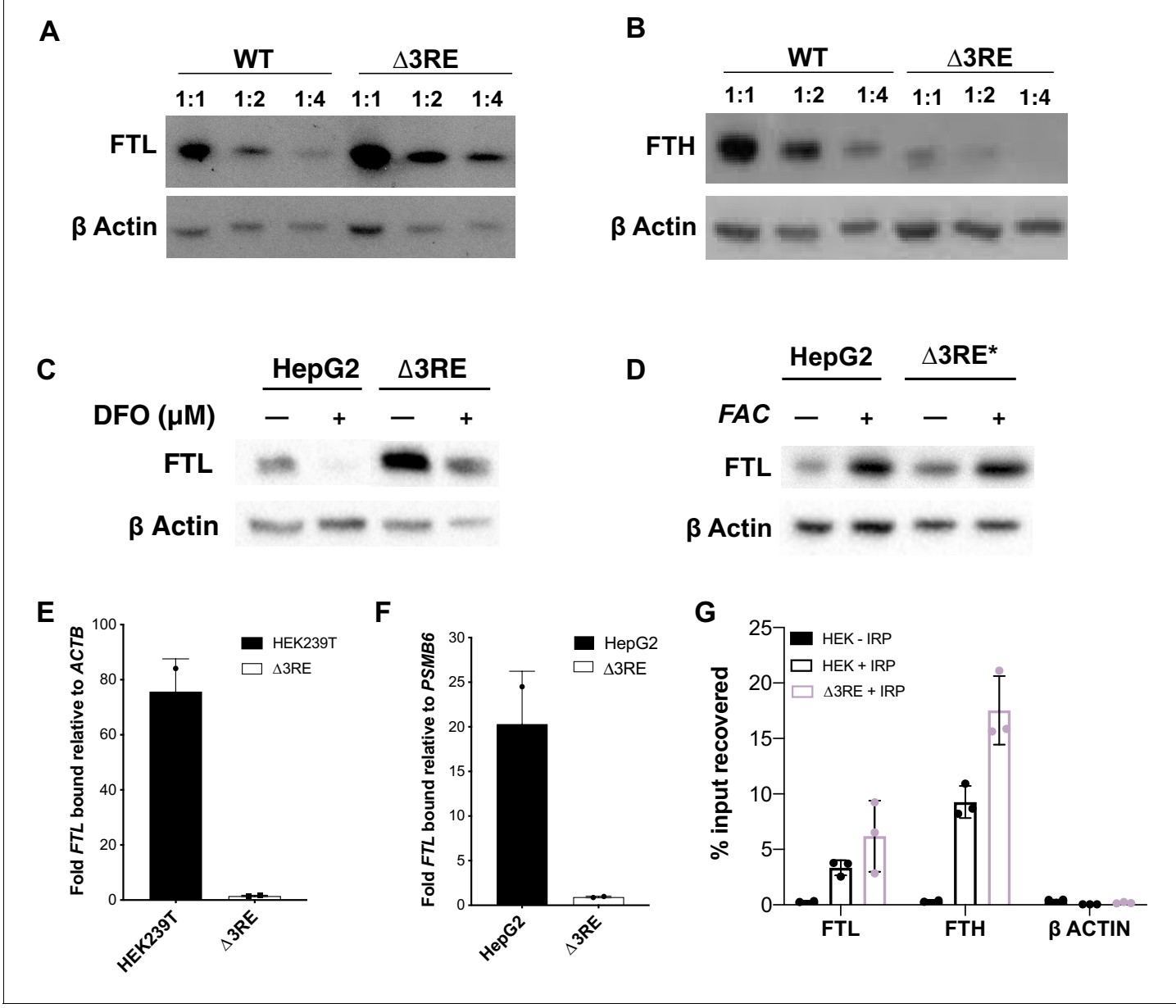

**Figure 3.** Physiological effects of the endogenous removal of the 3RE repressive element. (A,B) Representative western blots of (A) FTL and (B) FTH levels in the edited (Δ3RE) and WT HepG2 cells under normal iron conditions. Serial dilutions were used in order to better visualize the significance of the changes in FTL and FTH protein abundance. (C, D) Representative western blots of FTL levels in the edited (Δ3RE) and WT HepG2 cells under high- or low-iron conditions. Iron donor treatment with FAC at 50 μg/mL for 24 hr, and Iron chelation treatment with DFO at 50 μM for 48 hr. The asterisk (*) indicates that lysate from Δ3RE cells were diluted two-fold, due to the higher overall levels of FTL in these cells. All FTL blots are representative of three or more replicates. (E, F) Determination of the preferential binding of eIF3 to *FTL* mRNA via EIF3B immunoprecipitation (IP) followed by RNA extraction and RT-qPCR in both (E) HEK293T and (F) HepG2 cell lines. Control mRNAs used to normalize IPs were *PSMB6* and *ACTB*. Error bars represent the standard deviation of duplicate qPCR measurements from representative IP reactions. (G) Determination of IRP1 binding to *FTL* mRNA in WT (HEK + IRP) and *Δ3RE* (Δ3RE + IRP) HEK293T cells via FLAG immunoprecipitation (IP) followed by RNA extraction and RT-qPCR. The *ACTB* mRNA was used to control for nonspecific binding to FLAG-tagged IRP. HEK − IRP reflects cells that were not transiently transfected, but were carried through the IP and following experiments. Error bars represent the standard deviation for triplicate measurements from representative IP reactions.

DOI: https://doi.org/10.7554/eLife.48193.012

The following source data and figure supplements are available for figure 3:

**Source data 1.** Data anlysis of eIF3B and FLAG-tagged IRP1 immunoprecipitations.
DOI: https://doi.org/10.7554/eLife.48193.022

**Figure supplement 1.** CRISPR-Cas9 editing to remove the proposed eIF3-*FTL* interaction site.

*Figure 3 continued on next page*

*Figure 3 continued*

DOI: https://doi.org/10.7554/eLife.48193.013

**Figure supplement 2.** mRNA levels in edited cell lines.

DOI: https://doi.org/10.7554/eLife.48193.014

**Figure supplement 2—source data 1.** qPCR readouts relative to native*FTL*across various iron treament conditions.

DOI: https://doi.org/10.7554/eLife.48193.015

**Figure supplement 3.** Physiological effects of the endogenous removal of the 3RE on FTH protein levels.

DOI: https://doi.org/10.7554/eLife.48193.016

**Figure supplement 4.** Quantification of FTL and FTH protein levels in HepG2 cells.

DOI: https://doi.org/10.7554/eLife.48193.017

**Figure supplement 4—source data 1.** Western blot quantification of FTL and FTH in HepG2 cells.

DOI: https://doi.org/10.7554/eLife.48193.018

**Figure supplement 5.** Quantification of FTL and FTH protein levels in HEK293T cells.

DOI: https://doi.org/10.7554/eLife.48193.019

**Figure supplement 5—source data 1.** Western blot quantification of FTL and FTH in HEK293T cells.

DOI: https://doi.org/10.7554/eLife.48193.020

**Figure supplement 6.** Analysis of the ferritin complex upon deletion of the 3RE in FTL mRNA.

DOI: https://doi.org/10.7554/eLife.48193.021

*figure supplement 4D*, *Figure 3—figure supplement 5B*), with basal FTH levels reduced in the *Δ3RE* compared to WT cells.

To further ensure that IRP-mediated repression was maintained in the *Δ3RE* cell lines, we transiently transfected the WT and *Δ3RE* cell lines with plasmids encoding C-terminally FLAG-tagged IRP1. We then used FLAG immunoprecipitation followed by qPCR to determine whether IRP is bound to the edited *FTL* and other IRE containing mRNAs in vivo. We found that FLAG-tagged IRP bound *FTL* mRNA similarly in the wild type and *Δ3RE* cell line (*Figure 3G*). Intriguingly, *FTH1* mRNA was recovered at a considerably higher level in the *Δ3RE* cell line compared to wild type cells (*Figure 3G*), suggesting increased IRP binding. This increased binding of IRP to *FTH1* mRNA may explain the concurrent decrease in FTH abundance seen in *Figure 3B*. Notably, this increase in IRP binding to *FTH1* mRNA does not appear to be a simple mass action effect, as ferroportin levels–also regulated by an IRP-IRE interaction (*Anderson and McLaren, 2012*)–are not altered in the Δ3RE cell line (*Figure 3—figure supplement 5C*). Taken together, these data further support the hypothesis that the observed de-repression in the *Δ3RE* cells is due to the modulation of eIF3-based regulation of *FTL* translation, and not due to disruption of IRP-mediated regulation.

We proceeded to investigate whether the lack of eIF3-based repression of *FTL* translation and concomitant decrease in FTH protein levels had any effects on the assembled ferritin complexes. We purified the ferritin complexes from either wild-type HepG2 cells or the *Δ3RE* cell line using a fractional methanol precipitation protocol (*Cham et al., 1985*). We observed that the ferritin complex in the *Δ3RE* cell line was far more stable than that isolated from wild-type cells. Without ferric ammonium citrate (FAC) treatment to stabilize the complex (*Linder, 2013*), ferritin purified from WT cells consistently degraded, unlike the stable complexes from the *Δ3RE* cell line (*Figure 3—figure supplement 6*). This implicates eIF3 in regulating the dynamics and stability of the ferritin complex, as FTL-enriched ferritin complexes have been shown to be more stable under a wide array of denaturing condition (*Santambrogio et al., 1992*). Taken together, these results support the hypothesis that removal of the eIF3 interaction site in the 5'-UTR of the *FTL* mRNA derepresses *FTL* translation and can have a dramatic effect on ferritin subunit composition in the cell.

## SNPs in *FTL* that cause hyperferritinemia

Although the *Δ3RE* mutation in *FTL* revealed eIF3-dependent repression of *FTL* translation, it is not clear what role eIF3 may play in ferritin homeostasis in humans. The human genetic disease hereditary hyperferritinemia cataract syndrome (HHCS) is an autosomal dominant condition that primarily results in early onset of cataracts due to ferritin amassing in the lens (*Millonig et al., 2010*). HHCS arises from SNPs or deletions in the 5'-UTR of *FTL*, which are thought to disrupt the IRP-IRE interaction leading to increased *FTL* translation. For example, mutations observed in the apical loop of the IRE directly disrupt critical contacts essential for IRP binding to the IRE (*Figure 4A*) (5). Interestingly,

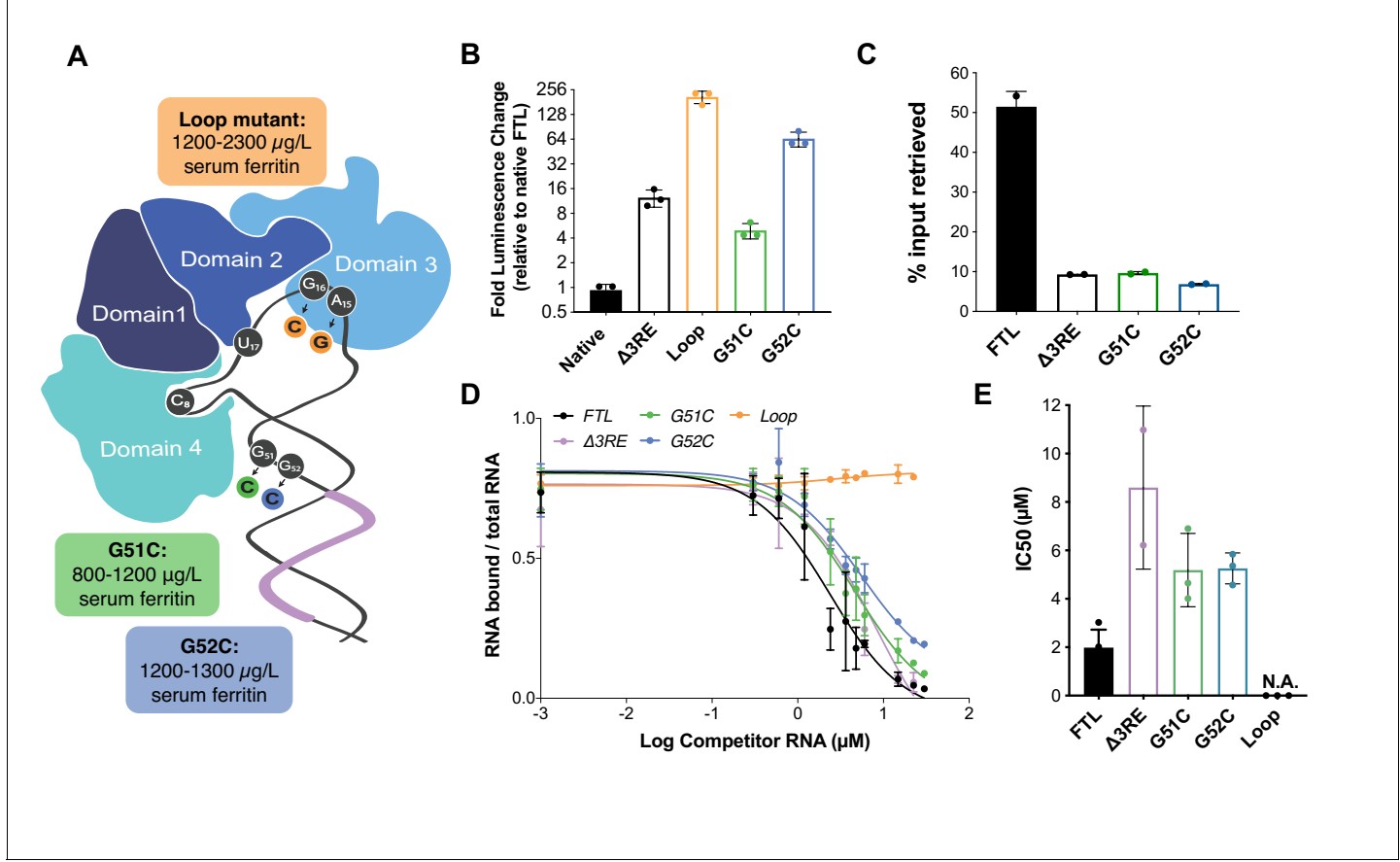

**Figure 4.** Role of eIF3 in select cases of hyperferritinemia. (**A**) Diagram of IRP binding to the IRE in *FTL* mRNA (*Anderson and McLaren, 2012*). Hyperferritinemia mutations are highlighted in orange (*Cazzola et al., 1997*), green (*Camaschella et al., 2000*), and blue (*Luscieti et al., 2013*) with their corresponding serum ferritin levels listed. Normal serum ferritin levels are under 300 μg/L. The 3RE is indicated in purple. Nucleotides that directly interact with the IRP are also identified (i.e. A15, G16, U17). (**B**) Luciferase activity of HepG2 cells transfected with mRNAs encoding the WT *FTL* 5'-UTR or various hyperferritinemia mutations (G51C, G52C, or Loop mutant (A15G/G16C)), normalized to WT *FTL* reporter luciferase luminescence. The results are from three biological replicates, with error bars representing the standard deviation of the mean. (**C**) Binding of eIF3 to luciferase reporter mRNAs with WT or mutant forms of the *FTL* 5'-UTR, using EIF3B immunoprecipitation (IP), followed by RNA extraction and RT-qPCR. Cells were harvested 8 hr post-transfection. Data are shown as the percent in the IP, compared to input levels. Error bars are the standard deviation of the mean of duplicate qPCR measurements from a representative IP. (**D**) Dose-response curve of RNA competition assays, based on gel shifts of NIR-labeled WT IRE-containing RNA, with WT, G51C, G52C, or Loop mutant (A15G/G16C) RNAs serving as cold competitors. Fold excess of competitor WT extended up to 100,000x. Recombinant IRP1 was used. Error bars represent standard deviations for each concentration of competitor. (**E**) The calculated IC$_{50}$ values of the various competitor RNAs, based on the data in (D), with error bars representing the standard deviation from the mean IC$_{50}$ value. N.A., the IC$_{50}$ value could not be determined for the Loop mutant due to lack of any detectable competition. Note that the data for Δ3RE in panels (D) and (E) are from *Figure 2B and C*, measured in duplicate. The remaining experiments in (D) and (E) were carried out in triplicate.

DOI: https://doi.org/10.7554/eLife.48193.023

The following source data and figure supplements are available for figure 4:

**Source data 1.** Luciferase reporter readouts, eIF3B Immunoprecipitation quantification, and EMSA analysis for hyperferritinemia mutants.
DOI: https://doi.org/10.7554/eLife.48193.027

**Figure supplement 1.** Native gels resolving RNA-IRP1 complexes formed after competition experiments with hyperferritinemia-related RNAs.
DOI: https://doi.org/10.7554/eLife.48193.024

**Figure supplement 2.** Iron responsiveness of hyperferritinemia associated SNPs in the *FTL* 5'-UTR.
DOI: https://doi.org/10.7554/eLife.48193.025

**Figure supplement 2—source data 1.** Luciferase reporter readouts and data analysis of FLAG-tagged IRP1 immunoprecipitations.
DOI: https://doi.org/10.7554/eLife.48193.026

two SNPs identified in certain patients with hyperferritinemia, G51C and G52C, disrupt the nucleotides one and two bases upstream of the annotated eIF3 PAR-CLIP site (*Figure 4A*) (*Camaschella et al., 2000*),(*Luscieti et al., 2013*). Although the PAR-CLIP methodology maps the region of interaction between an RNA and protein of interest, it does not always capture the full interaction site due to the requirement for 4-thiouridine cross-linking and subsequent RNase digestion to generate fragments for deep sequencing (*Ascano et al., 2012*), (*Hafner et al., 2010*). Thus, we wondered whether the G51C and G52C SNPs could potentially impact eIF3 repression of *FTL* mRNA translation, due to the proximity of the G51C and G52C mutations to the eIF3 PAR-CLIP site. We generated luciferase reporter constructs with either of the G51C and G52C SNPs, as well as a control with the previously described mutations in the IRE apical loop, and used mRNA transfections to test their effects on luciferase translation levels. All mutations led to an increase in luciferase levels, indicating that they alleviated translational repression (*Figure 4B*). We also observed that the G51C and G52C mRNAs do not interact with eIF3, based on eIF3 immunoprecipitations from transfected HEK293T cells (*Figure 4C*). Furthermore, the G51C and G52C SNPs maintained near wild-type IRP binding (*Figure 4D and E*), in stark contrast to the mutations in the IRE apical loop (*Figure 4A*), which completely abolished the interaction between IRP1 and the 5'-UTR element (*Figure 4—figure supplement 1*). Furthermore, using stable cell lines, we observe iron-responsive regulation of luciferase reporter mRNAs with G51C or G52C SNP-containing *FTL* 5'-UTRs, as well as IRP binding to these mRNAs (*Figure 4—figure supplement 2*). These results identify SNPs in *FTL* that cause hyperferritinemia likely due to disruption of eIF3-dependent repression of *FTL* translation.

## Discussion

We have shown that eIF3 represses the translation of *FTL* mRNA by binding a region of the *FTL* 5'-UTR immediately adjacent to the IRE. Upon disruption of eIF3 binding, the basal level of FTL production increases dramatically without affecting the iron-responsiveness of *FTL* translation (*Figure 3*), or binding of IRP to the remainder of the 5'-UTR containing the IRE (*Figure 2*, *Figure 3*, *Figure 4*). Taken together, these results expand the classical model of *FTL* mRNA post-transcriptional regulation by the iron-responsive IRP/IRE interaction to include a functionally distinct eIF3-dependent repressive mechanism (*Figure 5*). The physiological need for a dual repressive system involving IRPs and eIF3 in normal human health remains to be determined. Experiments combining the *Δ3RE* and apical loop mutations (*Figure 2F*, *Figure 2—figure supplement 3*) suggest that eIF3 and IRP can act simultaneously to repress *FTL* translation. Due to the close proximity of the IRE and eIF3 binding sites, and as suggested by our mathematical modeling (*Figure 2—figure supplement 3*), it may be possible that eIF3 physically interacts with IRP when they bind to the 5'-UTR in *FTL* mRNA in certain tissue or cellular contexts. Although we have identified a role for eIF3 in *FTL* mRNA translation regulation, it is still unclear what role eIF3 may play in response to iron level modulation, a key question to answer in the future. It is possible that both eIF3 and IRP are required for the proper iron responsiveness of *FTL* translation.

   Our present findings also provide the first molecular evidence for the direct involvement of eIF3 in a human disease caused by SNPs in the human population. We found that disruption of eIF3-mediated regulation of *FTL* translation could serve as the dominant cause of certain cases of hyperferritinemia. The specific genetic mutations we analyzed (G51C, G52C) map to the less-conserved lower stem of the IRE, and are predicted to have no direct physical interaction with IRP1 (PDB: 3SNP) (*Walden et al., 2006*). Here, we establish that these mutations minimally interfere with IRP's interaction with the IRE, whereas they greatly impact the eIF3-based interaction and *FTL* translational repression (*Figure 4C–E*). This work highlights how even highly clustered SNPs can contribute to disease through divergent molecular mechanisms. In the case of *FTL*, these clustered SNPs can disrupt three different aspects of translation regulation: IRP binding, eIF3 binding, or RNA folding (*Figure 5*).

   While our results connect eIF3 translational control to specific examples of hyperferritinemia, they also suggest a broader role for eIF3 in other hyperferritinemias and ferritin-related diseases. For example, we observed that derepression of *FTL* translation by disruption of the eIF3 interaction site leads to a concomitant decrease in FTH levels, driving an overall increase in ferritin complex stability (*Figure 3—figure supplement 6*). Ferritin is a known mediator of inflammatory responses, raising the question of whether eIF3 may contribute to ferritin's role in inflammation (*Morikawa et al.,*

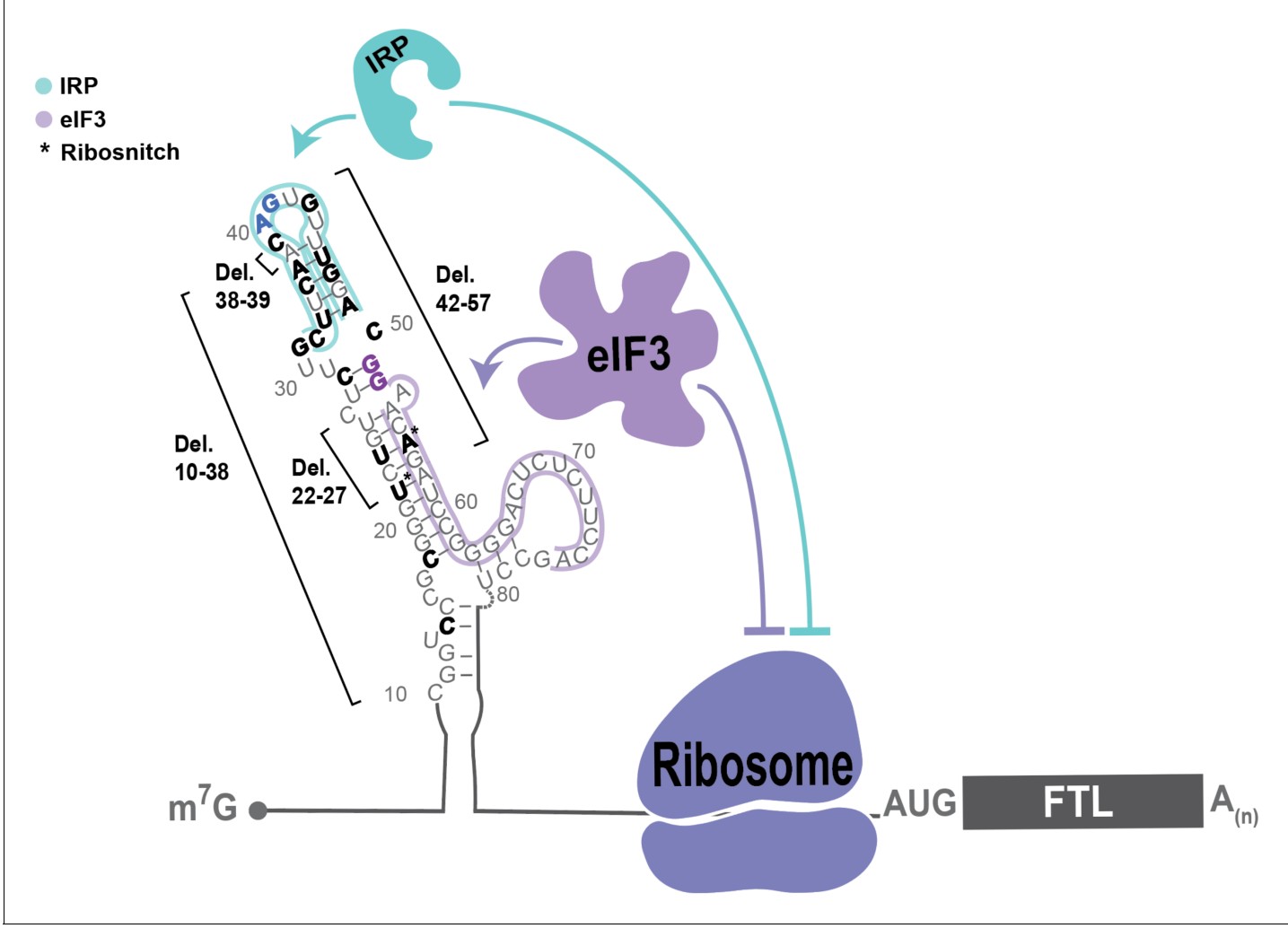

**Figure 5.** Model of post-transcriptional regulation of *FTL* mRNA. IRPs repress *FTL* mRNA translation in an iron-dependent manner, whereas eIF3 represses *FTL* translation in an iron-independent manner. Coordination between IRP repression and eIF3 repression may differ by cell and tissue context. Various hyperferritinemia mutations (bolded) listed in the literature are mapped on the experimentall -determined secondary structure of the *FTL* mRNA 5'-UTR (*Martin et al., 2012*), (*Luscieti et al., 2013*). The minimal annotation of the IRE is denoted by with a blue outline and the eIF3 PAR-CLIP defined interaction site is denoted with a purple outline. Mutations that disrupt IRP binding used in this study and determined here to disrupt eIF3 binding are bolded in blue and purple, respectively. (*) indicates nucleotides identified as ribosnitches (*Martin et al., 2012*).
DOI: https://doi.org/10.7554/eLife.48193.028

1995),(*Recalcati et al., 2008*). Our results provide new insights that should help connect molecular mechanisms of translational control to disease-associated SNPs identified in ever expanding genomic databases.

## Materials and methods

### Plasmids

The *FTL* 5'-UTR was amplified from human cDNA, and cloned into the pcDNA4 vector with a modified Kozak sequence (*Kranzusch et al., 2014*), by using In-Fusion HD Cloning Kit (Takara, Cat.# 638911) to generate the starting luciferase reporter plasmids. The *FTL* transcription start site is derived from the FANTOM5 database (*Lizio et al., 2015*). Additional mutations were generated through around-the-horn cloning using either blunt primers for deleting regions or primers with overhangs to introduce single or double nucleotide mutations. The IRP1 protein expression plasmid

was generated by amplifying the human IRP1 sequence from human cDNA and inserting it into the 2B-T vector (Addgene, plasmid # 29666) following a His$_6$ tag and TEV protease cleavage site. The IRP-FLAG construct was generated by inserting a 1X FLAG tag at the C-terminal end of IRP in the pCMV6-XL4 backbone (OriGENE, SC126974).

## In vitro transcription

RNAs were transcribed using T7 RNA polymerase prepared in-house. For luciferase reporter mRNAs, 5'-capping and 3'-polyadenylation were performed co-transcriptionally by including 3′-O-Me-m$^7$G (5′)ppp(5′)G RNA Cap Structure Analog (NEB, Cat.# S1411L) and using linearized plasmid template that had a sequence encoding a poly-A tail. Non-labeled RNAs for the IRP1 electrophoresis mobility shift assays were generated in the same manner, except the templates were not polyadenylated. Additionally, RNAs with the 38-nucleotide extension between the 5' -cap and IRE were constructed using a random nucleotide sequence. The exact nucleotide composition 5' of the IRE was previously reported to not significantly impact IRP binding (*Goossen and Hentze, 1992*). RNAs were purified after DNA template digestion by phenol-chloroform extraction and ethanol precipitation.

For genome editing, we used tandem CRISPR-Cas9 enzymes programmed with single-guide RNAs (sgRNAs) targeting the *FTL* gene, along with a single-stranded DNA (ssDNA) oligonucleotide homologous to the regions spanning the deleted 3RE sequence (*Figure 3—figure supplement 1*). sgRNAs were designed using the CRISPR.MIT.EDU program from the Feng Zhang Lab, MIT. CRISPR-Cas9-sgRNA was assembled as RNA-protein complexes (RNPs) (*Kim et al., 2014*). The DNA for transcription was synthesized by appending the sgRNA sequence downstream of a T7 RNA polymerase promoter. The DNA was then purified using phenol-chloroform extraction followed by iso-propanol precipitation. After transcription, the RNA products were treated with DNase I (Promega, Cat.# M6101), run on a 10% denaturing polyacrylamide gel (6 M urea), and extracted from the gel using the crush and soak method and ethanol precipitation.

## Luciferase reporter transfections

Human HepG2 cells were maintained in DMEM (Invitrogen 11995–073) with 10% FBS (Seradigm) and 1% Pen/Strep (Gibco, Cat.# 15140122). Transfections of the luciferase reporter mRNAs were done using the Mirus *Trans*IT-mRNA Transfection Kit (Cat.# MIR 2250), with the following protocol modifications. The day prior to transfection, HepG2 cells were seeded into opaque 96-well plates so that they would reach 80% confluence at the time of transfection. At this point, 200 ng of 5'-capped and 3'-polyadenylated mRNA was added at room temperature to OptiMEM media (Invitrogen, Cat. # 31985–088) to a final volume of 10 μL. All mRNA concentrations were determined using a Nano-drop, and were validated by running samples on an agarose gel. Bands on the gel were quantified to properly ensure all constructs were transfected at equal amounts. Then, 0.6 μL of both Boost reagent and TransIT mRNA reagent were added to the reaction and left to incubate at room temperature for 3 min. The TransIT-mRNA Reagent:mRNA Boost:RNA complex was distributed to the cells in a drop wise manner. Luciferase activity was assayed either 6 hr, or 12 hr post-transfection using the *Renilla* Luciferase assay kit (Promega, Cat.# E2820) and a Microplate Luminometer (Veritas). We used 6 hr incubation times for these experiments in order to keep readouts for various mutant constructs in a linear range (*Bert et al., 2006*). The only transfection not shown at 6 hr (*Figure 4B*) was also carried out for 6 hr in *Figure 2—figure supplement 3*. In all experiments, we define biological replicates as cells cultured in separate wells, and technical replicates as multiple measurements from cells from the same well.

In order to monitor the stability of transfected mRNAs during the timecourse of the luciferase reporter experiments, these 750 ng of each mRNA was transfected into HEK293T cells at 80% con-fluency well in a 24-well plate. After 6 hr, the cells were collected, pelleted, and then the RNA was extracted using the Direct-zol RNA Mini prep kit (R2051). cDNA was the prepared using 250 ng of RNA and Superscript IV (Thermo Fisher scientific, Cat. # 18091050). qPCR was carried out with the primers to the luciferase coding sequence with run conditions as described below in the methods for determination of transcript level abundance. We note that a completely accurate quantification of accessible mRNAs in the cell is limited due to technical complications inherent with mRNA trans-fection protocols and mechanisms (*Kirschman et al., 2017*).

## Cell line generation

Cell lines (HEK293T and HepG2) were obtained from the University of California Berkeley Cell Culture Facility, validated by STR analysis, and confirmed to be mycoplasma-free by the facility. To generate *FTL*-eIF3 interaction site null cell lines (Δ3RE), we used tandem CRISPR-Cas9 enzymes programmed with single-guide RNAs (sgRNAs) targeting the *FTL* gene, along with a single-stranded DNA (ssDNA) donor with homology to the regions spanning the deleted Δ3RE sequence. The sgRNAs were generated as described above, and targeted regions on both sides of the eIF3 interaction site (*Figure 2A*). The RNP complex was generated by incubating 100 pmol Cas9 with the two sgRNAs at a 1:1.2 Cas9 to total sgRNA ratio. This mixture was heated to 37℃ for 10 min and then kept at room temperature until use. The ssDNA donor was 90 nucleotides long, with 45-nucleotide homology on either side of the predicted double-strand cut sites allowing it to have perfect homology to the predicted edited sequence.

The Cas9-sgRNA RNP complexes, along with 500 pmol of the ssDNA donor, were transfected into either $5 \times 10^5$ HEK293T or HepG2 cells using the Lonza 96-well shuttle system SF kit (Cat. # V4SC-2096). The nucleofection programs used were as follows: CM-130 for HEK293T and EH-100 for HepG2. The transfected cells were left to incubate for either 48 hr for HEK293T cells or 72 hr for HepG2 cells before harvesting and extracting gDNA using QuickExtract (Epicentre: QE09060). The editing efficiency of individual Cas9-sgRNA RNPs was determined using a T7 endonuclease one assay (*Reyon et al., 2012*). The efficiency of the dual-sgRNA editing approach was determined by PCR-amplifying a 180-base pair region around the eIF3 interaction site, and analyzing the resulting products on a 6% non-denaturing 29:1 polyacrylamide gel. This method achieved an editing efficiency of nearly 100% in HEK293T cells and roughly 85% in HepG2 cells (*Figure 3—figure supplement 1B*). Monoclonal populations of edited cells were sorted using FACS, screened, and the final edited sequence was determined using TOPO TA cloning (*Ramlee et al., 2015*).

## Transcript level abundance

Total transcript abundance was determined by lysing $1.25 \times 10^6$ cells with Qiazol lysis buffer followed by using the Directzol RNA extraction kit (Zymo Research, Cat. # R2061), according to the manufacturer's instructions. The cDNA was generated by reverse transcription using 350 ng of RNA, random hexamers, and Superscript IV (Thermo Fisher scientific, Cat. # 18091050). Primers for the qPCR were as follows: FTL forward: 5′ -ACCTCTCTCTGGGCTTCTAT-3′, FTL reverse: 5′ -AGCTGGCTTCTTGATGTCCT-3′ (*Cozzi et al., 2004*), ACTB forward: 5′-CTCTTCCAGCCTTCCTTCCT-3′, ACTB reverse: 5′-AGCACTGTGTTGGCGTACAG-3′ (*Chen et al., 2008*), PSMB6 forward: 5′-GGACTCCA-GAACAACCACTG-3′, PSMB6 reverse: 5′-CAGCTGAGCCTGAGCGACA-3′ (*Mokany et al., 2013*), FTH1 forward: 5′-CGCCAGAACTACCACCA-3′, FTH1 reverse: 5′-TTCAAAGCCACATCATCG-3′ (*Liu et al., 2013*), 18S forward: 5′-GGCCCTGTAATTGGAATGAGTC-3′, 18S reverse: 5′-CCAAGA TCCAACTACGAGCTT-3′ (*Lee et al., 2016*), RLUC forward: 5′-GGAATTATAATGCTTATCTACG TGC-3′, RLUC reverse: 5′-CTTGCGAAAAATGAAGACCTTTTAC-3′ (*Kong et al., 2008*). Run conditions were: 95 ℃ for 15 s, followed by 40 cycles of 95 ℃ for 15 s, 60 ℃ for 60 s, 95 ℃ for 1 s.

## RNA immunoprecipitation and qPCR

The EIF3B-RNA immunoprecipitations were adapted from *Ramlee et al. (2015)* with the following modifications. One 15 cm plate of either HEK293 or HepG2 cells was used to prepare cell lysate using a NP40 lysis buffer (50 mM HEPES-KOH pH 7.5, 150 mM KCl, 2 mM EDTA, 0.5% Nonidet P-40 alternative, 0.5 mM DTT, 1 Complete EDTA-free Proteinase Inhibitor Cocktail tablet per 50 mL of buffer). The lysate was precleared with 15 μL of Dynabeads preloaded with rabbit IgG (Cell Signaling, Cat. # 2729) for one hour at 4 ℃. The lysate was collected and then incubated with a fresh 15 μL aliquot of Dynabeads and 7.5 μL of anti-EIF3B antibody (Bethyl A301-761A) for two hours at 4 ℃. Preparation of cDNA and qPCR primers are described and listed above.

For EIF3B immunoprecipitations of transfected mRNAs, 2.15 μg of mRNA was transfected into 1 well of either HEK293T or HepG2 cells in a 12-well plate using the protocol above. Cells were then left to incubate for 8 hr before harvesting. The cells were lysed using the NP40 lysis buffer listed above (*Lee et al., 2015*), and precleared with 2 μL of rabbit IgG-coated Dynabeads for one hour at 4 ℃. The lysate was collected and then incubated with a fresh 2 μL aliquot of Dynabeads and 4 μL of anti-EIF3B antibody (Bethyl A301-761A) for 2 hr at 4 ℃. RNA was collected, cDNA prepared and

qPCR carried out with the primers and run conditions as described in the methods for transcript level abundance.

For FLAG-tagged IRP1 immunoprecipitations, 2.2 µg of plasmid DNA was transfected into a 10 cm dish of 80% confluent HEK239T WT cells or HEK293T $\Delta 3RE$ mutant cells. Cells were then left to incubate for 24 hr before harvesting. The cells were lysed using the NP40 lysis buffer as described in *Ramlee et al. (2015)* and then further diluted 3x with the lysis buffer lacking DTT. The lysate was collected and then incubated with pre-equilibrated anti-FLAG antibody conjugated agarose beads (Sigma A2220) for two hours at 4 ˚C. The beads were then washed with a high salt wash buffer (50 mM HEPES-KOH pH 7.5, 300 mM KCl, 2 mM EDTA, 1% Nonidet P-40 alternative) three times. The protein was eluted from the beads using two washes with 1X FLAG peptide for 30 min each at 4 ˚C. The RNA was collected using phenol/chloroform extraction followed by ethanol precipitation. cDNA was prepared using Superscript IV (Thermo Fisher scientific, Cat. # 18091050) and qPCR carried out with the primers and run conditions as described above in the methods for determination of transcript level abundance.

## Iron level modulation

In order to modulate the iron levels in cells, HepG2 and HEK293T cells were treated with a final concentration of either 50 µg/mL of ferric ammonium citrate (FAC) (an iron donor) for 24 hr or either 200 µM for 24 hr or 50 µM for 48 hr of desferoxamin (DFO) (an iron chelator) before harvesting.

For the iron treatment of cells with the stably integrated luciferase reporters harboring the FTL 5'UTR SNPs, either $H_2O$, 200 µM of DFO, or 50 ug/mL of FAC was added to an individual 96 well of 80% confluent cells. The cells were allowed to incubate for 24 hr before taking the luminescence reading. BioRender was used for the cell schematic in *Figure 4—figure supplement 2A*.

## IRP1 purification

IRP1 was purified based on the protocols in *Carvalho and Meneghini (2008)*, *Basilion et al. (1994)* with modifications. The IRP1-encoding 2B-T plasmid was transformed into chemically-competent BL21 Rosetta pLysS *E. coli*, using heat shock at 42 ˚C, and grown on Ampicillin plates. A single colony was used to inoculate a 5 mL LB culture containing Ampicillin, which was then used to inoculate a 50 mL starter culture that was allowed to reach saturation overnight. Approximately $4 \times 10$ mL of the overnight culture was used to inoculate $4 \times 1$L cultures using ZY5052 media lacking the 1000x trace metal mix (30 mM HEPES, 5% glycerol, 43 mM KCl, 0.5 mM EDTA, 0.5 mM DTT) (*Studier, 2005*) plus Carbomicillin. The 1 L cultures were grown at 37 ˚C to $OD_{600}$ = 0.36, at which point the temperature was lowered to 18 ˚C, and allowed to grow at 18 ˚C for 36 hr prior to harvest.

Pelleted *E. coli* cells were lysed using sonication in lysis buffer (30 mM HEPES pH = 7.5, 400 mM KCl, 5% Glycerol and 1 mM DTT) along with Protease inhibitor (Roche, Cat. # 5056489001) tablets. Lysate was loaded on a 5 mL Ni-NTA pre-packed HiTrap column (GE, Cat. # 17-5248-02), allowed to incubate at 4 ˚C for 1 hr, before eluting using 600 mM imidazole in the same buffer as above. Pooled fractions from the elution were then dialyzed overnight into ion-exchange (IEX) buffer (30 mM HEPES pH 7.5, 1 mM DTT, 5% Glycerol, and 1 mM EDTA), for subsequent purification using a 5 mL HiTrap Q-column (GE, Cat. # 17-1154-01). Samples were then loaded on a Q column using IEX buffer, and the column was washed with eight column volumes of IEX buffer without KCl. IRP1 was eluted using 800 mM KCl in IEX buffer.

## IRP1 electrophoresis mobility shift assays

To detect IRP1 binding by native gel shifts, RNA samples were transcribed using the Atto-680 RNA Labeling Kit (Jena Bioscience, FP-220–680) and subsequently purified using RNA Clean and Concentrator-25 columns (Zymo, R1018). This form of labeling has been shown not to disrupt protein-RNA interactions (*Köhn et al., 2010*). Unlabeled RNA was transcribed and purified as described above.

Binding experiments were carried out with a final concentration of 300 pM of labeled RNA and 225 nM of recombinant human IRP1, which facilitated a 1:1 ratio of RNA binding to IRP1. We first ensured the RNA competition experiments reached equilibrium – which required at least 11 hr of incubation – by measuring the approximate dissociation rate constant ($k_{off}$) of WT *FTL* 5'-UTR from IRP1. Heparin was included at the beginning of the reaction at a final concentration of 4.5 µg/mL. The initial binding reaction was carried out in a 1x RXN buffer (40 mM KCl, 20 mM Tris-HCL, pH 7.4,

2 mM MgCl$_2$, 2 mM DTT, and 5% Glycerol) for 30 min at room temperature (*Goforth et al., 2010*), (*Fillebeen et al., 2014*). For competition experiments, unlabeled RNA was then added in concentrations 1000x-100,000x that of the labeled RNA. In preliminary experiments, we found the k$_{off}$ to be roughly 0.006 min$^{-1}$ using an 8 hr incubation time course with competitor. We then tested the fraction of IRP bound after 11 hr and 18 hr incubations with competitor *FTL* RNA and observed no changes in the residual fraction of IRP bound to RNA (~15%), indicating the reactions had reached equilibrium. Thus, subsequent experiments with competitor RNAs were carried out for 18 hr, after the first 30 min pre-incubation in the absence of competitor. The reactions were resolved on Tris-glycine gels (Thermo Fisher Scientific, Cat.# XP04122BOX) and gels was visualized using an Odyssey Licor set to 700 nM wavelength and an intensity of 6.5. Band intensity quantification was carried out using Image Studio (Licor). The IC$_{50}$ values for each competitor RNA was determined using Graph Pad Prism 7 (Graph Pad Software) from a set of triplicate experiments, except for the Δ3RE competitor RNA, which was tested in duplicate.

## Ferritin complex purification

The ferritin complex purification procedure was adapted from *Cham et al. (1985)*, with slight modifications. Either one 15 cm dish of Δ3RE cells grown in normal media or eight 15 cm dishes of wild-type HepG2 cells that had been treated with 50 ng/mL FAC for 24 hr were harvested, weighed, and lysed in 4x weight/volume NP40 lysis buffer (50 mM HEPES-KOH pH = 7.5, 150 mM KCl, 2 mM EDTA, 0.5% Nonidet P-40 alternative, 0.5 mM DTT, 1 Complete EDTA-free Protease Inhibitor Cocktail tablet per 10 mL of buffer). Samples were incubated on ice for ten minutes and then centrifuged for 10 min at 21,000xg at 4 ˚C. Samples were diluted in 1:2 phosphate buffered saline (PBS), and methanol was added to the diluted lysate to a final concentration of 40% (v/v). The sample was then heated to 75 ˚C for 10 min. After cooling on ice for 5 min, the samples were centrifuged in a microfuge 20R at 1251xg RPM for 15 min at 4 ˚C. The resulting supernatant was collected and concentrated using a 100 k MW cutoff Amicon filter (Cat.# UFC510024) by centrifugation for 10 min at 14000xg. The sample was washed once with PBS and spun for an additional 4 min at 14000xg. The sample was then collected by inverting the column and centrifuging the sample for 2 min at 1000xg at 4 ˚C. All samples collected were brought to a final volume of 80 µL with PBS. The purity of the sample was determined by running the sample on a native gel (Thermo Fisher Scientific, Cat.# XP04200BOX) followed by Coomassie staining.

## Western blots

The following antibodies were used for Western blot analysis: anti-EIF3B (Bethyl A301-761A) at 1:1000; anti-FTL (Abcam, ab69090) at 1:800; anti-FTH (Santa Cruz, sc-25617) at 1:400; anti-IRP1 (Abbexa, abx004618) at 1:400; anti-IRP2 (Abcam, ab129069) at 1:800; and anti-b-Actin (Abcam, ab8227) at 1:1000, anti-Ferroportin (Novus, NBP 1–2150255) at 1:300.

## Lentiviral transduction

The G51C and G52C Renilla luciferase reporters were cloned into the NLV103 plasmid. Virus was generated using LentiX cells in a 10 cm dish format and TransIT-LT1 transfection reagent. Virus was harvested and filtered after 48 and 72 hr. Total virus was pooled and 500 µL of fresh virus was added to 10$^6$ HEK293T cells along with 10 µg/µL of polybrene (Millipore). Cells were left to incubate for 48 hr before a 4 day selection process with 4 µg/mL of puromycin. Cells were split in non-selective media once before use.

## Mathematical modeling of IRP and eIF3 co-occupancy on *FTL* mRNA

We tested a mathematical model in which IRP and eIF3 do not bind simultaneously to the same *FTL* mRNA. This model thus includes three possible states of the *FTL* mRNA: the fraction bound solely by IRP ($x_1$), the fraction bound solely by eIF3 ($x_2$), and the remainder of the mRNA, which is unbound by either factor ($x_3$) (Figure 2 – figure supplement 2). The model also assumes that the mutations introduced into *FTL* mRNA do not affect the translational efficiency of unbound mRNA species, that is translation of $x_3$ is identical for all 4 mRNAs. The translational efficiency of IRP-bound mRNA ($y_1$) and eIF3-bound mRNA ($y_2$) ranges from fully repressed ($y = 0$) to completely unbound and derepressed ($y = 1$). Thus, the translational efficiency for the bound populations ($x_1$ and $x_2$) is less than 1,

while the translational efficiency of $x_3$ is equal to 1. We also include the parameters $\alpha$ and $\beta$ to account for changes in the distribution, or shifts, of previously bound mRNA ($x_2$ and $x_1$) to new populations ($0 \leq \alpha + \beta \leq 1$). Taken together, four equations present the luciferase output of each *FTL* mRNA:

$$\text{FTL} = y_1 x_1 + y_2 x_2 + x_3 \tag{1}$$

$$\Delta 3\text{RE} = y_1(x_1 + \alpha) + x_2 - \alpha + x_3 \tag{2}$$

$$\text{Loop} = x_1 - \beta + y_2(x_2 + \beta) + x_3 \tag{3}$$

$$\text{Double} = x_1 + x_2 + x_3 \tag{4}$$

Here, FTL represents the luciferase readout of wild-type *FTL* mRNA; Δ3RE the luciferase readout of *FTL* mRNA with the Δ3RE mutation; Loop, the luciferase readout of *FTL* mRNA with the IRE loop mutation; and Double the luciferase readout of FTL mRNA with both the Δ3RE and Loop mutations. In order to solve this system of equations, we proceeded to use experimentally determined values as seen in *Figure 2F*. We assume the Δ3RE and Loop mutations disrupt regulation by the respective factor (IRP or eIF3) completely, consistent with the biochemical results in *Figure 4*. In these cases, both $y_1$ and $y_2$ revert to a value of 1, that is the same translational efficiency of $x_3$.

To reduce the number of variables, we rearranged *Equation (1)* with normalized luciferase values ($x_3 = 1 - y_1 x_1 - y_2 x_2$) and substituted it into *Equations (2)* through (4)

$$\Delta 3\text{RE} = y_1(x1 + \alpha) + x_2 - \alpha + 1 - y_1 x_1 - y_2 x_2 \rightarrow 1 + y_1 \alpha - \alpha + (1 - y_2)x_2 \tag{5}$$

$$\text{Loop} = x_1 - \beta + y_2(x_1 + \beta) + 1 - y_1 x_1 - y_2 x_2 \rightarrow 1 + y_2 \beta - \beta + (1 - y_1)x_1 \tag{6}$$

$$\text{Double} = x_1 + x_2 + 1 - y_1 x_1 - y_2 x_2 \rightarrow 1 + (1 - y_1)x_1 + (1 - y_2)x_2 \tag{7}$$

Further rearrangement and substitution of Δ3RE and Loop into *Equation (7)* yields:

$$(1 - y_2)x_2 = \Delta 3RE - 1 - y_1 \alpha + \alpha \tag{8}$$

$$(1 - y_1)x_1 = Loop - 1 - y_2 \beta + \beta \tag{9}$$

$$Double = Loop + \Delta 3RE + (\alpha + \beta - y_1 \alpha - y_2 \beta - 1) \tag{10}$$

Given the measured luciferase values (*Figure 2F*), the above model, which assumes IRP and eIF3 do not bind simultaneously to the same *FTL* mRNA, is inconsistent with the data, even accounting for measurement error.

Referring to *Equation (10)*:

$$41.8(\pm 6.5) > 12.5(\pm 2.9) + 4.4(\pm 0.4) + (\alpha + \beta - y_1 \alpha - y_2 \beta - 1)$$

$$41.8(\pm 6.5) > 16.9(\pm 2.9) + (\alpha + \beta - y_1 \alpha - y_2 \beta - 1)$$

Thus, IRP and eIF3 likely can act in cis on *FTL* mRNAs.

## Acknowledgements

We are immensely grateful to Bruno Martinez for providing us with purified IRP1, Luisa Arake De Tacca for help optimizing the eIF3 immunoprecipitations and for many constructive discussions, Hector Nolla for help with flow cytometry and single cell sorting, and Alison Killilea and the Berkeley tissue culture facility for cells and advice.

## Additional information

### Funding

| Funder | Grant reference number | Author |
|---|---|---|
| National Institute of General Medical Sciences | P50 GM102706 | Mia C Pulos-Holmes<br>Daniel N Srole<br>Maria G Juarez<br>Amy S-Y Lee<br>David Trombley McSwiggen<br>Nicholas T Ingolia<br>Jamie H Cate |
| National Institute of General Medical Sciences | R01 GM065050 | Mia C Pulos-Holmes<br>Daniel N Srole<br>Maria G Juarez<br>Jamie H Cate |
| American Heart Association | 16PRE30140013 | Mia C Pulos-Holmes |

The funders had no role in study design, data collection and interpretation, or the decision to submit the work for publication.

### Author contributions

Mia C Pulos-Holmes, Conceptualization, Formal analysis, Supervision, Funding acquisition, Validation, Investigation, Visualization, Methodology, Writing—original draft, Project administration, Writing—review and editing; Daniel N Srole, Investigation, Writing—review and editing, assisted in carrying out experiments related to screening and obtaining clonal cell lines, as well as plasmid construct generation; Maria G Juarez, Investigation, Writing—review and editing, aided in plasmid construct generation and accessory biochemical experiments; Amy S-Y Lee, David T McSwiggen, Conceptualization, Methodology, Writing—review and editing; Nicholas T Ingolia, Formal analysis, Supervision, Methodology, Writing—review and editing; Jamie H Cate, Conceptualization, Formal analysis, Supervision, Funding acquisition, Visualization, Methodology, Writing—original draft, Project administration, Writing—review and editing

### Author ORCIDs

Mia C Pulos-Holmes https://orcid.org/0000-0002-8234-4020
David T McSwiggen https://orcid.org/0000-0003-3844-7433
Nicholas T Ingolia http://orcid.org/0000-0002-3395-1545
Jamie H Cate https://orcid.org/0000-0001-5965-7902

### Decision letter and Author response

Decision letter https://doi.org/10.7554/eLife.48193.031
Author response https://doi.org/10.7554/eLife.48193.032

## Additional files

### Supplementary files

• Transparent reporting form
DOI: https://doi.org/10.7554/eLife.48193.029

### Data availability

All data generated or analysed during this study are included in the manuscript and supporting files.

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
