## [Decision Letter]

Thank you for submitting your article "Repression of ferritin light chain translation by human eIF3" for consideration by *eLife*. Your article has been reviewed by three peer reviewers, one of whom is a member of our Board of Reviewing Editors, and the evaluation has been overseen by James Manley as the Senior Editor. The following individual involved in review of your submission has agreed to reveal their identity: Leos Shivaya Valasek (Reviewer #2).

The reviewers have discussed the reviews with one another and the Reviewing Editor has drafted this decision to help you prepare a revised submission.

Summary:

This report provides evidence that eIF3 binds to ferritin light chain mRNA (FTL) adjacent to and overlapping with the IRE element where IRP1 binds to exert the well-known repression of FTL mRNA translation in low iron conditions. They show that a deletion of the eIF3 binding site that does not include the region of overlap with the IRE derepresses expression of a FTL-5'UTR-LUC reporter by ~10-fold (although the fold-increase seems to vary between 4x and 20X in different experiments). This Δ3RE deletion does not strongly reduce binding of recombinant IRP1 to the FTL 5'UTR in vitro, reducing affinity ~4-fold; and it also still confers derepression of reporter mRNA in the presence of mutations that impair IRP1 binding or function (although the extent of derepression by Δ3RE is diminished when IRP1 function is reduced by moving the IRE further from the cap.) Gene editing was used to introduce Δ3RE into FLT mRNA in cultured cells and it is shown that the mutation derepresses native FTL protein expression apparently without reducing native IRP1 binding to the mRNA (although there are several issues complicating this conclusion), and evidence is presented that the increased FTL expression and concomitant decreased FTH chain expression confers increased stability on the ferritin complex (although there are issues with these results too.) In addition, the expected responses of FTL expression to iron excess or depletion achieved pharmacologically remain intact for the Δ3RE mutated FTL gene. These findings lead the authors to conclude that eIF3 represses translation of FTL mRNA independently of IRP1. They go on to show that two SNPs in FTL associated with hyperferritinemia that are expected to affect IRP1 binding but are close to the eIF3 binding site that they mapped previously by PAR-CliP derepress expression of the FTL-5'UTR-LUC reporter comparably to the Δ3RE mutation and reduce binding of eIF3 to the reporter mRNA in cells, but only weakly reduce binding of recombinant IRP1 to the FTL mRNA in vitro. It was not shown however that the SNP mutations allow for normal IRP1 binding to reporter mRNA in cells; nor was it shown that the mutated reporters respond normally to iron excess or depletion, which appear to be a significant omissions. The results are significant in linking eIF3-mediated repression of a specific mRNA to a human disease and also add an additional layer of regulation to the well-established mechanism of iron-mediated translational control by IRP1.

Essential revisions:

– It should be verified that equal amounts of WT vs delta3RE and WT vs. G51C and G52C reporter mRNAs are present in the transfected cells from which the luciferase expression levels were obtained

– Results shown in panels 2E and 2F appear to be inconsistent. In 2E, delta3RE causes a 32-fold luminescence change; in 2F, the same mutation/construct causes only a 4.4-fold change. Moreover, in 2F, IRP and eIF3 binding have an “additive” (rather than “synergistic”) effect, whereas in 2E, there is no such additive effect (the expectation would be more than 32-fold). The authors need to convincingly address this issue, ideally by performing both sets of experiments in parallel under identical conditions (according to legend 2F, e.g. the incubation times were different).

– Explain the need for the mathematical modelling in Figure 2—figure supplement 2 and the modelling itself in much greater detail in the Results section. The authors may wish to consider if at all, the modelling adds significantly to the scientific quality of the paper.

– Figure 3—figure supplement 1D-E should be presented in the main figures

– Regarding Figure 3, for panels A and D, the results of Western blots from multiple biological replicates must be presented with appropriate loading controls, and statistical analysis of the differences in mean values determined. Also, for panels A and C, a complete set of WB and RNA-Ip experiments from the same WT and mutant cell lines must be presented.

– Discuss a possible mechanism for how delta3RE leads to increased IRP binding to FTH1 mRNA

– The results in Figure 3—figure supplement 4 do not provide convincing evidence that the WT and mutant ferritin complexes differ in stability. The authors should either show increased ferritin stability from delta3RE cells by thermal melting of purified complexes using urea or guanidinium chloride or cite convincing literature where increasing the FTL/FTH ratio was shown to increase ferritin stability.

– It should be shown by analyzing the existing reporters containing the G51C and G52C mutations that these mutations leave intact IRP1 mediated repression of reporter expression in response to changing iron levels achieved using FAC and DFO treatments.

– Make the appropriate revisions of text to address all other major comments of the reviewers.

*Reviewer #1:*

This report is of interest in providing evidence that eIF3 binds to ferritin light chain mRNA (FTL) adjacent to and overlapping with the IRE element where IRP1 binds to exert the well-known repression of FTL mRNA translation in low iron conditions. They show that a deletion of the eIF3 binding site that does not include the region of overlap with the IRE derepresses expression of a FTL-5'UTR-LUC reporter by ~10-fold (although the fold-increase seems to vary between 4x and 20X in different experiments). This Δ3RE deletion does not strongly reduce binding of recombinant IRP1 to the FTL 5'UTR in vitro, reducing affinity ~4-fold; and it also still confers derepression of reporter mRNA in the presence of mutations that impair IRP1 binding or function (although the extent of derepression by Δ3RE is diminished when IRP1 function is reduced by moving the IRE further from the cap.) Gene editing was used to introduce Δ3RE into FLT mRNA in cultured cells and it is shown that the mutation derepresses native FTL protein expression apparently without reducing native IRP1 binding to the mRNA (although there are several issues complicating this conclusion), and evidence is presented that the increased FTL expression and concomitant decreased FTH chain expression confers increased stability on the ferritin complex (although there are issues with these results too.) In addition, the expected responses of FTL expression to iron excess or depletion achieved pharmacologically remain intact for the Δ3RE mutated FTL gene. These findings lead the authors to conclude that eIF3 represses translation of FTL mRNA independently of IRP1. They go on to show that two SNPs in FTL associated with hyperferritinemia that are expected to affect IRP1 binding but are close to the eIF3 binding site that they mapped previously by PAR-CliP derepress expression of the FTL-5'UTR-LUC reporter comparably to the Δ3RE mutation and reduce binding of eIF3 to the reporter mRNA in cells, but only weakly reduce binding of recombinant IRP1 to the FTL mRNA in vitro. It was not shown however that the SNP mutations allow for normal IRP1 binding and translational repression by IRP1 in cells, and it seems important to show at least that the mutated reporters respond normally to iron excess or depletion. The results are significant in linking eIF3-mediated repression of a specific mRNA to a human disease and also add an additional layer of regulation to the well-established mechanism of iron-mediated translational control by IRP1. However, there are a number of significant shortcomings with the study, as noted above, and described below.

Major comments:

– Figure 1E: Determine the effects of the Δ3RE mutations on reporter mRNA levels, and show that Δ3RE increases the polysome size of the reporter mRNA in the event that it is found to reduce mRNA levels, in order to rigorously establish derepression at the level of translation.

– Figure 3A: the results of Western blots from multiple biological replicates must be presented and statistical analysis of the differences in mean values determined to rigorously establish that the Δ3RE mutation derepresses FTL protein expression.

– Subsection “Physiological response to loss of eIF3-dependent repression” and Figure 3A and C: Why weren't WB results from HEK cells presented, whereas the text would lead one to believe that the effect of the mutation was the same in both cell lines? Also, why were the RNA-Ips shown in panel C done only in HEK cells and not HepG2 cells, as these data are not directly relevant to the WB expression data in panel A? This selective presentation of results is troubling, and a complete set of WB and RNA-Ip experiments from the same WT and mutant cell lines should be presented.

– Figure 3D: It's unclear from the extensive cropping of the data that the results derive from matched WT and Δ3RE cells examined in parallel. In addition, the results from multiple biological replicates must be presented and statistical analysis of the differences in mean fold- derepression or fold-repression values determined between WT and Δ3RE cells.

– Figure 3—figure supplement 4: It's not immediately clear that the smearing attributed to degradation of the WT ferritin complex would not be seen for the mutant complex if more of the latter was loaded on the gel for equal comparison with WT. The amount of intact complex versus degraded species from multiple biological replicates must be presented and statistical analysis of the differences in mean values of the ratios determined.

– Figure 4E: It is necessary to determine the effects of the G51C and G52C mutations on reporter mRNA levels, and show that they increase the polysome size of the reporter mRNA if they are found to reduce mRNA levels, in order to rigorously establish derepression at the level of translation.

– Subsection “SNPs in FTL that cause hyperferritinemia”: they have not shown that the G51C and G52C mutations leave intact IRP1 mediated repression in response to changing iron levels. This should be done at the very least for the reporters using FAC and DFO treatment.

*Reviewer #2:*

In the presented paper, Pulos-Holmes et al. explore the role of eIF3 in the translational control of the ferritin light chain (FTL) gene, the deregulation of which causes hyperferritinemia. They propose that eIF3 acts as a specific repressor of the FTL mRNA translation that binds to its structured 5' UTR simultaneously with another repressor – the iron regulatory protein (IRP) – and thus adds another, IRP-IRE-independent layer of the negative control. Since a few specific SNPs located in the eIF3-binding site in the FTL 5' UTR that have been associated with the aforementioned disease disturb this eIF3-specific control, authors conclude that this is the first study identifying a direct role for eIF3-mediated translational control in a specific human disease.

It is an interesting and well-executed work that, in my opinion, deserves to be eventually published in the *eLife* journal.

Major criticism:

The IRE is a well-defined structural element that has been specifically mutated and I think that this work would benefit from the analysis of secondary structures of the deltaPAR and delta3RE mutant mRNAs; it may greatly help with the interpretation of the proposed synergy between IRP and eIF3. Also, I did not understand why the delta3RE deletion goes beyond the PAR-CLIP 3' boundary (nt 76) further downstream to nt 90, when compared to deltaPAR. This was not rationalized.

Were the Luc experiments (like the one shown in Figure 1E) normalized to the mRNA levels? Can the authors rule out that the presence of IRE partially destabilizes FTL mRNA and both tested mutations, highly likely eliminating its structure, at least partially elevate its expression by stabilizing the mRNA?

Figure 2E and F. Taking into account the synergy (additivity) observed in Figure 2F, wouldn't one expect the same in Figure 2E; i.e. a higher bar for delta3RE with the extended form? Also, the magnitude of the repression relieve by this construct differs dramatically between native forms in Figure 1E (10-fold) and 2E (30-fold), why would that be?

Figure 3—figure supplement 1D-E should be presented in the main figure, it is important in my opinion.

Figure 3C. How could delta3RE lead to an increase of IRP binding to the eIF3-independent mRNA of FTH1?

Subsection “Physiological response to loss of eIF3-dependent repression” final paragraph: This implicates eIF3 in regulating not only the abundance of the two ferritin subunits (How could it regulate FTH? Please discuss.), but also the dynamics and stability of the ferritin complex." How could this happen? IRP should be doing the same, shouldn't it? Please discuss.

Figure 4C. I guess it would be informative to examine the binding of the loop mutant to eIF3 as well.

The authors should at least try to discuss what could be the physiological importance of the IRP-IRE-independent eIF3-mediated control of the FTL expression to lift the perception that this study is more descriptive than innovative.

*Reviewer #3:*

This is a timely and important study by the Cate lab and collaborators. It examines at the molecular level whether and how the general eukaryotic translation initiation factor 3 (eIF3) affects the translation of a specific target mRNAs (human ferritin light chain mRNA, FTL mRNA) to which it makes sequence or structure-specific contacts. This work is a follow-up of a previous genome-wide PAR-CLIP analysis by the Cate lab that identified numerous eIF3 target mRNAs. Pulos-Holmes now demonstrate as a proof-of-principle how particular disease-causing human SNPs in the 5'UTR of FTL mRNA cause translational mis-regulation as a consequence of failed eIF3 interaction and not, as previously assumed, as a consequence of a failed interaction with the iron-response-protein (IRP). IRP is a highly specific RNA-binding protein with a close-by but distinct RNA target site and long known to regulate FTL mRNA in an iron-dependent manner. Conceptually, the present work advances our understanding of how also general translation factors such as eIF3 can cause specific RNA regulation in addition to and in combination with specific RNA-binding proteins. Furthermore, the authors identify the molecular pathway of how particular SNPs cause human disease and they show that even closely-spaced SNPs can act via distinct molecular interactions.

The manuscript is concise, generally well-written and easy to follow also for a general audience. Nevertheless, I have some comments and suggestions, also regarding experimental details, that should be clarified before publication.

1) It should be useful to add nucleotide numbers to Figure 1D and Figure 1—figure supplement 1C, also marking already here the delta3RE and loop dinucleotide mutations (“loop”) that are used throughout the work. Also the legends should be updated accordingly. Otherwise it is for example not quite clear that “loop” means point mutations and not a larger deletion.

2) Also please explain the meaning of “Double” in the legend of Figure 2. What does the line “N.A., the IC50 value.…” in the Figure 2C legend refer to?

3) The Figure referring to the “mathematical modelling” (Figure 2—figure supplement 2) is not very well explained. What is y1 and y2, what are the brown balls, what is meant by “previously”? One can understand the idea of the modelling only after reading through the Materials and methods in detail. Modeling is claimed to “demonstrate that IRP and eIF3 seem to work synergistically”. This sentence sounds odd. In my view, the described modeling approach seems like an “overkill”, unnecessarily complicated and rather indirect, because the data in the end excludes the described model. Isn't it sufficient in Figure 2F to see that, starting from “Double”, the repression in “FTL” (~40-fold) is roughly the product of the repression by IRP in “delta3RE” (~10 fold) and the repression by eIF3 in “Loop” (~4-fold)? If only one of the two binders could act at a time, the maximum repression would be (~10 fold) by IRP. Maybe I am missing something, but then it needs to be better explained.

4) The description of the Western blots and sample dilutions in Figure 3A, B, D and Figure 3—figure supplement 3 is not very clear, and/or it may be advisable to show more convincing blots. It is not always clear which lane shows diluted samples and whether the β Actin controls were co-diluted or not (It seems to me that the b Actin controls were run on separate lanes of the gel and were undiluted). Moreover, in Figure 3D, the lanes in the absence of FAC would be expected to look exactly like the lanes in the absence of DFO (in both cases the cells are still untreated), but this is not really the case. Although all of the cells are still reactive to FAC or DFO, it would be more convincing if the starting material looked the same. Similar in Figure 3—figure supplement 3.

5) In Figure 3B, it might be good to clarify that “delta3RE” refers to the respective mutation in the FTL RNA, e.g. by labelling “delta3RE” in Figure 3A, B as “FTL-delta3RE”. Moreover, the authors do not give any explanation for the puzzling result that FTH protein expression is apparently decreased when FTL RNA is mutated. Is it possible that “delta3RE” cells try to compensate the lack of FTL RNA translational repression by producing generally more IRP protein and that this excessive IRP protein then downregulates FTH RNA? Did the authors try and look at endogenous IRP protein levels in WT and “delta3RE” cells with e.g. a suitable antibody?

6) Is there any explanation for why DFO does not downregulate FTH RNA in Figure 3—figure supplement 2B?

7) It should be clarified in Figure 3—figure supplement 4 that this is a Coomassie-stained native gel (maybe show in blue?) of the purified ferritin complex, presumably containing different ratios of FTL and FTH chains (predominantly FTL in the case of delta3RE cells?). The text claims that the delta3RE cells contain 8x as much ferritin as the WT cells, but this is not obvious from the figure. Moreover, the text also claims the complexes from delta3RE cells to be more “stable”. Presumably this does not refer to the physiological half-life of the complex in the cell but to the “stability” of the complex in the harsh purification procedure using 40% methanol and 75 deg C. Does the ferritin complex remain intact and folded in this process or does it undergo unfolding/refolding and/or disassembly/reassembly? Does protein concentration and purity matter in this process and possibly lead to wrong conclusions regarding “stability”?

8) The text lacks general information for a general reader on what is the physiological difference between FTL and FTH proteins and whether the ratio in which they occur in ferritin has physiological importance. Are FTL-exclusive ferritins more stable than FTH-containing ones? This is important, because eIF3 binding appears to affect not only the total amount of FTL, but even more so the ratio of FTL to FTH, possibly generating ferritin complexes with different properties. This is somewhat mentioned at the end of the Discussion, but the authors could elaborate on this point of physiological significance.

9) Some statements in the text could be revisited:

Abstract: SNPs.… “are thought” to disrupt.…. Do you mean “were described/reported” to disrupt? Otherwise that statement sounds like a hypothesis and creates the impression that the study might continue with a proof of IRP-dependence. See also the first paragraph of subsection “SNPs in FTL that cause hyperferritinemia” for a very similar statement.

Subsection “Identification of the eIF3-FTL mRNA interaction site”: “a secondary structure element that overlaps” or do you mean “a 24 nucleotide sequence.… that overlaps”?

In the same section: “.… in a deletion the/that maintained”.…

Subsection “Decoupling the repressive role of eIF3 on FTL mRNA from that of IRP”: What is “the characteristic C bulge”?

In the same section: List/ explain the loop mutation as a A15G/G16C dinucleotide mutation (clarify numbers in Figures; 40/41 in Figure 5?)

Subsection “Physiological response to loss of eIF3-dependent repression”: “significantly” suggests statistically evaluated significance, maybe use “considerably”.

In the same section check the following statement: “This implicates eIF3 in regulating not only the abundance of the two ferritin subunits, but also the dynamics and stability of the ferritin complex” This statement needs further explanation on how this is thought to occur – for example that different ratios of FTL and FTH within a given ferritin complex could cause such differences (maybe add references if data exists on this).

Figure 3—figure supplement 2 legend: Explain FAC and DFO here instead of later in the Figure 3—figure supplement 3.

Figure 4 legend: small caps for “hyperferritinemia”. “EIF3B” or “eIF3B”? “in the IP compared to/with input levels”. “Loop” mutant with reference to Cazzola et al.? In 4D, clarify that this is competition for IRP-binding. Serum ferritin levels are given for loop mutants, but what is “normal”/physiological? Explain details of the mutants already in 4B, rather than in 4D. If available, add a “Loop”' column in 4C.

---

## [Author Response]

Essential revisions:– It should be verified that equal amounts of WT vs delta3RE and WT vs. G51C and G52C reporter mRNAs are present in the transfected cells from which the luciferase expression levels were obtained

All mRNA concentrations were determined using a Nanodrop and were validated by running samples on an agarose gel. Bands on the gel were quantified to properly ensure all constructs were transfected at equal amounts. We have clarified this approach in the Materials and methods.

In order to address the concern of RNA degradation, we performed a time course experiment in which we followed both the luminescence and mRNA amount (via qPCR) post-transfection. As shown in Figure 2—figure supplement 2, after 6 hrs posttransfection there is no significant difference in mRNA stability when comparing the different mRNA reporters. With this information we can now definitively conclude that the derepression we see occurs at the level of translation.

– Results shown in panels 2E and 2F appear to be inconsistent. In 2E, delta3RE causes a 32-fold luminescence change; in 2F, the same mutation/construct causes only a 4.4-fold change. Moreover, in 2F, IRP and eIF3 binding have an “additive” (rather than “synergistic”) effect, whereas in 2E, there is no such additive effect (the expectation would be more than 32-fold). The authors need to convincingly address this issue, ideally by performing both sets of experiments in parallel under identical conditions (according to legend 2F, e.g. the incubation times were different).

There are two potential factors that could explain why the derepression values observed in Figure 2F are seemingly not consistent with the extended ∆3RE construct in Figure 2E. First, the extension of the 5′-UTR from its native state does not completely abolish the repression mediated by IRP. It only significantly dampens it (See Goossen and Hentze, 1992). This is in contrast to the loop mutations which abolish IRP binding entirely (Figure 4D and Cazzola et al., 1997). We note that in the original submission, the experiment in Figure 2E was carried out for 12 hrs. We have seen that with HepG2 cells there is still an increase in luminescence beyond shorter (i.e. 6 hr) incubation times. For example, see Figure 1—figure supplement 1D. In a separate experiment, which we now include as the new Figure 2E, we used a 6-hr incubation. In this experiment, derepression of the WT FTL with extended 5′-UTR is about 5-fold. By contrast, the loop mutation derepressed FTL translation 12-fold in the 6-hr incubation in Figure 2F.

Second, it is not clear how the distance from the 5′-cap affects repression mediated eIF3. This would be an interesting mechanistic question to address in a follow-up publication.

– Explain the need for the mathematical modelling in Figure 2—figure supplement 2 and the modelling itself in much greater detail in the Results section. The authors may wish to consider if at all, the modelling adds significantly to the scientific quality of the paper.

Reviewer 3 states that from the data presented in Figure 2F, “If only one of the two binders could act at a time, the maximum repression would be (~10 fold) by IRP,” not the ~40x that we see. However, this assumes that the WT *FTL* mRNA is fully repressed in these culture conditions. Based on the DFO experiments (i.e. in Figure 3C and Figure 3—figure supplement 4), this is most certainly not the case. This is one of the reasons we developed the model. We think the quantitative modeling provides a framework for understanding *FTL* translation regulation more generally. For example, in this manuscript we cannot rule out that eIF3 may have a role in iron responsiveness of *FTL* translation. By having this model present, we can provide possible explanations for why some of the iron responsiveness data is not an exact match between WT and edited cell lines.

– Figure 3—figure supplement 1D-E should be presented in the main figures

These data are now included in Figure 3E-F.

– Regarding Figure 3, for panels A and D, the results of Western blots from multiple biological replicates must be presented with appropriate loading controls, and statistical analysis of the differences in mean values determined. Also, for panels A and C, a complete set of WB and RNA-Ip experiments from the same WT and mutant cell lines must be presented.

The purpose of these experiments was to show that the edited cells retain the ability to respond to iron, which they do. Though quantification of these westerns is not necessary to convey this result, we have included quantification of these data in Figure 3—figure supplements 4 and 5. Figure 3C-D and Figure 3 —figure supplement 4 show a representative set of western blots and quantification of westerns carried out in multiple biological replicates. We have also included representative western blots and quantification of FTL and FTH levels in WT and ∆3RE HEK293T cells, in Figure 3 —figure supplement 5, which show similar results. We note that the increase in FTL abundance in the edited cell lines might alter the cellular response to iron treatment with respect to *FTL* translation. We therefore cannot expect a fully normal response in these edited cells.

With respect to the IRP-IP results, although we were also interested in preforming the corresponding IRP-FLAG IPs in HepG2 cells, we were only able to achieve efficient plasmid transfections in HEK293T cells. Multiple attempts and various forms of DNA transfection methods were attempted in HepG2 cells, with no success in our hands.

– Discuss a possible mechanism for how delta3RE leads to increased IRP binding to FTH1 mRNA.

This has been included in the main text. Due to the significant increase in FTL protein levels, we initially hypothesized that there is a feedback mechanism causing IRP to bind to *FTH1* mRNA to control the overall amount of ferritin there is in the cell. We do show that in the ∆3RE cell lines that IRP seems to bind more *FTH1* mRNA when compared to native cells using an IRP-FLAG IP (See Figure 3G and Figure 3—figure supplements 4 and 5). Although we have not checked if the abundance of IRP differs in these cell lines, if it did, other IRP-regulated transcripts such as that for Ferroportin would likely be affected in the same way as FTH. Thus, we have also looked at another protein, Ferroportin, which also has an IRE in the 5′-UTR of its mRNA. Notably, we see no difference in Ferroportin abundance comparing the wild type and ∆3RE cell line, suggesting that the decrease in FTH levels is not due to a mass action process. The mechanism by which IRP would be recruited preferentially to *FTH1* mRNA is beyond the scope of this paper.

– The results in Figure 3—figure supplement 4 do not provide convincing evidence that the WT and mutant ferritin complexes differ in stability. The authors should either show increased ferritin stability from delta3RE cells by thermal melting of purified complexes using urea or guanidinium chloride or cite convincing literature where increasing the FTL/FTH ratio was shown to increase ferritin stability.

The precedence for this stems from the fact FTL has a salt bridge in its central hydrophilic region that contributes to its stability (See Santambrogio et al., 1991, which we now cite). We have modified the text to clarify this point.

– It should be shown by analyzing the existing reporters containing the G51C and G52C mutations that these mutations leave intact IRP1 mediated repression of reporter expression in response to changing iron levels achieved using FAC and DFO treatments.

To address whether the G51C and G52C mRNA constructs retain the ability to interact with IRP in vivo,we generated stable HEK293T lines expressing the G51C and G52C luciferase reporter mRNAs using lentiviral constructs. This was done in HEK293T cells because we also wanted to repeat the IRP-FLAG IPs in these cell lines. As shown in Figure 4—figure supplement 2, IRP can still interact with the mutant FTL 5′-UTR elements with the G51C or G52C mutations, as previously determined using the in vitro EMSAs (Figure 4 and Figure 4—figure supplement 1). Although FAC treatment does not seem to release IRP as efficiently from the G51C and G52C reporters compared to endogenous FTL, we suggest that these mutant constructs are more efficiently bound by IRP in cells than the WT, as seen in the IRP-FLAG IP results in Figure 4—figure supplement 2D, or due to possible post-translational regulation of FTL, which is not recapitulated using luciferase as a reporter.

– Make the appropriate revisions of text to address all other major comments of the reviewers.

Additional modifications to the text and figures are described below.

Reviewer #1:

Please see our overall responses above.

Reviewer #2:

[…] It is an interesting and well-executed work that, in my opinion, deserves to be eventually published in the eLife journal.Major criticism:The IRE is a well-defined structural element that has been specifically mutated and I think that this work would benefit from the analysis of secondary structures of the deltaPAR and delta3RE mutant mRNAs; it may greatly help with the interpretation of the proposed synergy between IRP and eIF3. Also, I did not understand why the delta3RE deletion goes beyond the PAR-CLIP 3' boundary (nt 76) further downstream to nt 90, when compared to deltaPAR. This was not rationalized.

While we agree that obtaining the secondary structure for the mutant constructs would be quite informative, we think these experiments would be better suited to a follow-up study. We think the functional data in Figure 2 and Figure 2—figure supplement 1 provide convincing data that at least the IRE component of the 5’-UTR retains is structure in the ∆3RE, but not ∆PAR, context.

The ∆3RE extends past the PAR-CLIP site mainly due to CRISPR-Cas9 editing limitations. Since HDR has extremely low efficiency in HepG2 cells, we decided this was the best way to approach the editing. It also achieves both the goals of disrupting eIF3 and maintaining the IRP interaction.

Were the Luc experiments (like the one shown in Figure 1E) normalized to the mRNA levels? Can the authors rule out that the presence of IRE partially destabilizes FTL mRNA and both tested mutations, highly likely eliminating its structure, at least partially elevate its expression by stabilizing the mRNA?Figure 2E and F. Taking into account the synergy (additivity) observed in Figure 2F, wouldn't one expect the same in Figure 2E; i.e. a higher bar for delta3RE with the extended form? Also, the magnitude of the repression relieve by this construct differs dramatically between native forms in Figure 1E (10-fold) and 2E (30-fold), why would that be?Figure 3—figure supplement 1D-E should be presented in the main figure, it is important in my opinion.Figure 3C. How could delta3RE lead to an increase of IRP binding to the eIF3-independent mRNA of FTH1?Subsection “Physiological response to loss of eIF3-dependent repression” final paragraph: This implicates eIF3 in regulating not only the abundance of the two ferritin subunits (How could it regulate FTH? Please discuss.), but also the dynamics and stability of the ferritin complex." How could this happen? IRP should be doing the same, shouldn't it? Please discuss.

Please see our overall responses above.

Figure 4C. I guess it would be informative to examine the binding of the loop mutant to eIF3 as well.

We think these experiments would be better suited to a follow-up study, and is now covered by our analysis of the model presented in Figure 2—figure supplement 3.

The authors should at least try to discuss what could be the physiological importance of the IRP-IRE-independent eIF3-mediated control of the FTL expression to lift the perception that this study is more descriptive than innovative.

Please see our overall response above. We noted possible broader implications for eIF3mediated repression of *FTL* translation in the Discussion, and think the molecular insights we have obtained stand on their own as an innovative contribution to understanding human translation regulation.

Reviewer #3:

The manuscript is concise, generally well-written and easy to follow also for a general audience. Nevertheless, I have some comments and suggestions, also regarding experimental details, that should be clarified before publication.1) It should be useful to add nucleotide numbers to Figure 1D and Figure 1—figure supplement 1C, also marking already here the delta3RE and loop dinucleotide mutations (“loop”) that are used throughout the work. Also the legends should be updated accordingly. Otherwise it is for example not quite clear that “loop” means point mutations and not a larger deletion.

Since this is a cartoon schematic, we did not number the nucleotides in this panel. We added the loop mutation information to Figure 2F, the first time they appear. We have also included numbering in Figure 4A, Figure 5, as well as in the figure legend to Figure 1—figure supplement 1.

2) Also please explain the meaning of “Double” in the legend of Figure 2. What does the line “N.A., the IC50 value…” in the Figure 2C legend refer to?

We have updated the figure legend to Figure 2, to make this clear. Double means a construct that contains the loop mutation that disrupts IRP binding combined with the ∆3RE mutation which disrupts eIF3 binding.

3) The Figure referring to the “mathematical modelling” (Figure 2—figure supplement 2) is not very well explained. What is y1 and y2, what are the brown balls, what is meant by “previously”? One can understand the idea of the modelling only after reading through the Materials and methods in detail. Modeling is claimed to “demonstrate that IRP and eIF3 seem to work synergistically”. This sentence sounds odd. In my view, the described modeling approach seems like an “overkill”, unnecessarily complicated and rather indirect, because the data in the end excludes the described model. Isn't it sufficient in Figure 2F to see that, starting from “Double”, the repression in “FTL” (~40-fold) is roughly the product of the repression by IRP in “delta3RE” (~10 fold) and the repression by eIF3 in “Loop” (~4-fold)? If only one of the two binders could act at a time, the maximum repression would be (~10 fold) by IRP. Maybe I am missing something, but then it needs to be better explained.4) The description of the Western blots and sample dilutions in Figure 3A, B, D and Figure 3—figure supplement 3 is not very clear, and/or it may be advisable to show more convincing blots. It is not always clear which lane shows diluted samples and whether the β Actin controls were co-diluted or not (It seems to me that the b Actin controls were run on separate lanes of the gel and were undiluted). Moreover, in Figure 3D, the lanes in the absence of FAC would be expected to look exactly like the lanes in the absence of DFO (in both cases the cells are still untreated), but this is not really the case. Although all of the cells are still reactive to FAC or DFO, it would be more convincing if the starting material looked the same. Similar in Figure 3—figure supplement 3.5) In Figure 3B, it might be good to clarify that “delta3RE” refers to the respective mutation in the FTL RNA, e.g. by labelling “delta3RE” in Figure 3A, B as “FTL-delta3RE”. Moreover, the authors do not give any explanation for the puzzling result that FTH protein expression is apparently decreased when FTL RNA is mutated. Is it possible that “delta3RE” cells try to compensate the lack of FTL RNA translational repression by producing generally more IRP protein and that this excessive IRP protein then downregulates FTH RNA? Did the authors try and look at endogenous IRP protein levels in WT and “delta3RE” cells with e.g. a suitable antibody?

Please see our overall responses above.

6) Is there any explanation for why DFO does not downregulate FTH RNA in Figure 3—figure supplement 2B?

Unfortunately, we do not have an explanation for this at this time.

7) It should be clarified in Figure 3—figure supplement 4 that this is a Coomassie-stained native gel (maybe show in blue?) of the purified ferritin complex, presumably containing different ratios of FTL and FTH chains (predominantly FTL in the case of delta3RE cells?). The text claims that the delta3RE cells contain 8x as much ferritin as the WT cells, but this is not obvious from the figure. Moreover, the text also claims the complexes from delta3RE cells to be more “stable”. Presumably this does not refer to the physiological half-life of the complex in the cell but to the “stability” of the complex in the harsh purification procedure using 40% methanol and 75 deg C. Does the ferritin complex remain intact and folded in this process or does it undergo unfolding/refolding and/or disassembly/reassembly? Does protein concentration and purity matter in this process and possibly lead to wrong conclusions regarding “stability”?

We have clarified the figure legend to address that this is a coomassie stained gel. You are correct that we are referring to the stability of the complex during the purification process and not its half life. Based on literature precedence (cited in the manuscript) we believe that this is an intact and folded complex throughout the purification process. This method of purification is very common for ferritin and because of how harsh it is, we do not feel extremely worried about contaminants in the protein.

8) The text lacks general information for a general reader on what is the physiological difference between FTL and FTH proteins and whether the ratio in which they occur in ferritin has physiological importance. Are FTL-exclusive ferritins more stable than FTH-containing ones? This is important, because eIF3 binding appears to affect not only the total amount of FTL, but even more so the ratio of FTL to FTH, possibly generating ferritin complexes with different properties. This is somewhat mentioned at the end of the Discussion, but the authors could elaborate on this point of physiological significance.

There are physiological differences between FTL and FTH. For example, FTH is the catalytic subunit of the two whereas FTL promotes complex stability and aids in the iron mineralization process. We expanded our description of FTH in various places, i.e. in terms of its effect on Ferritin stability, and the observation of post-transcriptional downregulation of FTH levels, but not Ferroportin levels, based on our new experiments. However, we have not strongly elaborated on this process because we feel that it goes beyond the scope of the paper.

9) Some statements in the text could be revisited:Abstract: SNPs.… “are thought” to disrupt.…. Do you mean “were described/reported” to disrupt? Otherwise that statement sounds like a hypothesis and creates the impression that the study might continue with a proof of IRP-dependence. See also the first paragraph of subsection “SNPs in FTL that cause hyperferritinemia” for a very similar statement.

This has been corrected.

Subsection “Identification of the eIF3-FTL mRNA interaction site”: “a secondary structure element that overlaps” or do you mean “a 24 nucleotide sequence.… that overlaps”?

This has been corrected.

In the same section: “… in a deletion the/that maintained”.…

This has been corrected. Thank you for finding the typo.

Subsection “Decoupling the repressive role of eIF3 on FTL mRNA from that of IRP”: What is “the characteristic C bulge”?

We have modified Figure 2D by adding an asterisk, and the associated figure legend to clarify where the characteristic C bulge is in the *FTL* 5’-UTR.

In the same section: List/ explain the loop mutation as a A15G/G16C dinucleotide mutation (clarify numbers in Figures; 40/41 in Figure 5?)

The numbers for the loop mutants A15G/G16C have been noted in the Figure 2F legend, as well as the legend for Figure 4. The numbering represents their space in the IRE sequence, not the full RNA. Unfortunately, there is a difference in the naming system of the hyperferritinemia mutants.

Subsection “Physiological response to loss of eIF3-dependent repression”: “significantly” suggests statistically evaluated significance, maybe use “considerably”.

Changed as suggested.

In the same section check the following statement: “This implicates eIF3 in regulating not only the abundance of the two ferritin subunits, but also the dynamics and stability of the ferritin complex” This statement needs further explanation on how this is thought to occur – for example that different ratios of FTL and FTH within a given ferritin complex could cause such differences (maybe add references if data exists on this).

Please see our overall response above.

Figure 3—figure supplement 2 legend: Explain FAC and DFO here instead of later in the Figure 3—figure supplement 3.

Done.

Figure 4 legend: small caps for “hyperferritinemia”. “EIF3B” or “eIF3B”? “in the IP compared to/with input levels”. “Loop” mutant with reference to Cazzola et al.? In 4D, clarify that this is competition for IRP-binding. Serum ferritin levels are given for loop mutants, but what is “normal”/physiological? Explain details of the mutants already in 4B, rather than in 4D. If available, add a “Loop”' column in 4C.

Hyperferritinemia – fixed.

EIF3B is the human naming convention.

‘IP, compared to input levels’.

Normal levels of ferritin now included in figure legend.

Loop mutations clarified.

Unfortunately, we do not have these data.